# Pharmacological reversal of synaptic and network pathology in human *MECP2*-KO neurons and cortical organoids

Cleber A Trujillo[1,*,†] , Jason W Adams[1,2,3,†], Priscilla D Negraes[1,4], Cassiano Carromeu[1,4], Leon Tejwani[1,¶], Allan Acab[1], Ben Tsuda[2,5], Charles A Thomas[1], Neha Sodhi[4], Katherine M Fichter[4], Sarah Romero[4], Fabian Zanella[4], Terrence J Sejnowski[5,6,7], Henning Ulrich[8] & Alysson R Muotri[1,3,**]

## Abstract

Duplication or deficiency of the X-linked *MECP2* gene reliably produces profound neurodevelopmental impairment. *MECP2* mutations are almost universally responsible for Rett syndrome (RTT), and particular mutations and cellular mosaicism of *MECP2* may underlie the spectrum of RTT symptomatic severity. No clinically approved treatments for RTT are currently available, but human pluripotent stem cell technology offers a platform to identify neuropathology and test candidate therapeutics. Using a strategic series of increasingly complex human stem cell-derived technologies, including human neurons, *MECP2*-mosaic neurospheres to model RTT female brain mosaicism, and cortical organoids, we identified synaptic dysregulation downstream from knockout of *MECP2* and screened select pharmacological compounds for their ability to treat this dysfunction. Two lead compounds, Nefiracetam and PHA 543613, specifically reversed *MECP2*-knockout cytologic neuropathology. The capacity of these compounds to reverse neuropathologic phenotypes and networks in human models supports clinical studies for neurodevelopmental disorders in which MeCP2 deficiency is the predominant etiology.

**Keywords** cortical organoids; drug discovery; *MECP2* mosaicism; neurodevelopmental disease modeling; stem cells
**Subject Categories** Genetics, Gene Therapy & Genetic Disease; Neuroscience

## Introduction

*MECP2* is an X-linked gene that encodes an epigenetic regulatory protein, methyl CpG binding protein-2 (MeCP2), that is critical for typical human brain development (Chahrour *et al*, 2008; Gonzales & LaSalle, 2010). In addition to an array of other neurodevelopmental disorders (Watson *et al*, 2001; Shibayama *et al*, 2004; Samaco *et al*, 2005), *MECP2* loss-of-function mutations are the most common etiology of RTT (Amir *et al*, 1999), a severe neurodevelopmental disorder clinically characterized by head growth deceleration, profound cognitive decline, regression of acquired abilities, and stereotypies in early infancy (Hagberg *et al*, 2002). Critically, these syndromic clinical features appear following a brief period of normal development (Hagberg *et al*, 2002). Although the precise neurobiological changes linking *MECP2* mutations to the RTT phenotype are unclear, human postmortem tissue from RTT females portrays reduced brain size, decreased dendritic arborization and spine formation, and reduced synapse numbers (Johnston *et al*, 2003; Gonzales & LaSalle, 2010). Moreover, studies in murine models of *Mecp2* deficiency implicate an accordant deficit in synaptic function, including decreased synaptic transmission and plasticity (Moretti *et al*, 2006; Zhou *et al*, 2006; Dani & Nelson, 2009). These findings suggest that synaptic dysfunction is a central pathology of MeCP2 deficiency and may be a viable treatment target (Johnston *et al*, 2003; Gonzales & LaSalle, 2010).

Therapeutically targeting pathways downstream from *MECP2* has been proposed as a RTT treatment strategy (Braat & Kooy, 2015; Pozzo-Miller *et al*, 2015; Katz *et al*, 2016; Benke & Möhler, 2018),

1 Department of Pediatrics/Rady Children's Hospital, Department of Cellular & Molecular Medicine, School of Medicine, University of California San Diego, La Jolla, CA, USA
2 Department of Neurosciences, School of Medicine, University of California San Diego, La Jolla, CA, USA
3 Center for Academic Research and Training in Anthropogeny, University of California San Diego, La Jolla, CA, USA
4 StemoniX Inc, Maple Grove, MN, USA
5 Computational Neurobiology Laboratory, Salk Institute for Biological Studies, La Jolla, CA, USA
6 Institute for Neural Computation, University of California San Diego, La Jolla, CA, USA
7 Division of Biological Sciences, University of California San Diego, La Jolla, CA, USA
8 Departamento de Bioquímica, Instituto de Química, Universidade de São Paulo, São Paulo, Brazil
*Corresponding author. Tel: +1 858 534 9320; E-mail: ctrujillo@ucsd.edu
**Corresponding author. Tel: +1 858 534 9320; E-mail: muotri@ucsd.edu
†These authors contributed equally to the work
¶Present address: Interdepartmental Neuroscience Program, Yale School of Medicine, New Haven, CT, USA

and the narrow window of typical development suggests intervention during this period to bolster synaptic function may be opportune to preserve clinical function and ameliorate subsequent decline. Development and discovery of therapeutics for RTT and other neurodevelopmental disorders have been challenged by limited opportunity to investigate disease pathogenesis in a human model. Advances in pluripotent stem cell (PSC) technology, including three-dimensional neural differentiation, offer a promising human-based platform to evaluate candidate therapeutics for neurodevelopmental disorders and hasten clinical translatability (Adams *et al*, 2019).

Here, we employed a series of increasingly complex human stem cell-derived models as a screening platform to identify pharmacological compounds capable of specifically improving the neurocytological deficits caused by knockout of *MECP2* (*MECP2*-KO) without affecting control neurons. Using our strategic pipeline, we demonstrated *MECP2*-KO-attributable synaptic pathology in genetic expression, morphology, and physiology and screened a series of drugs with synapse-relevant mechanisms of action for their ability to improve these alterations. Two currently available lead compounds, Nefiracetam and PHA 543613, exhibited potential to partially rescue the synaptic defects caused by MeCP2 deficiency and are viable candidates for clinical trial.

## Results

### *MECP2*-KO produces synaptic pathology in human neurons

To develop our human models of MeCP2 deficiency, human PSCs with *MECP2* exonic loss-of-function mutations (Q83X nonsense or K82 frameshift with familial and isogenic PSC controls, respectively) were differentiated into cortical neurons as previously described (Figs 1A and B and EV1A–I, and EV2A–C; Espuny-Camacho *et al*, 2013; Nageshappa *et al*, 2016). We first compared the morphologies between control and *MECP2*-KO neurons. Compared with controls, *MECP2*-KO neurons transfected with Syn1::GFP lentiviral vectors

had decreased soma areas and spine density ($P < 0.0001$). The spine-like protrusions in *MECP2*-KO neurons were less stable ($P < 0.01$), but there was no observable difference in their formation ($P = 0.26$), length ($P = 0.28$), or motility ($P = 0.17$; Fig EV2D and E), supporting previous descriptions of RTT neuronal pathology (Marchetto *et al*, 2010; Nguyen *et al*, 2012).

Because MeCP2 is an epigenetic regulator, we sought to identify changes in gene expression resulting from its deficiency. RT–qPCR array (Figs 1C and D, and EV3A and B) and single-cell RT–qPCR (Figs 1E–G and EV3C–K) analyses at timepoints throughout neuronal differentiation from neural progenitor cells (NPCs; day 0) revealed differences in genetic expression between *MECP2*-KO and control neurons. Concordant with the prominent roles of MeCP2 in synaptic function and neuronal maturation (Gonzales & LaSalle, 2010), many of the implicated genes appeared relevant to synaptic function and showed differences in expression during differentiation (Fig 1C–G). In further agreement with previous findings of RTT pathology (Stuss *et al*, 2012), our quantitation of cellular markers of neuronal fate and function in cells positive for the neuronal marker MAP2 revealed a decreased proportion of layer V and VI cortical neurons (CTIP2[+] and FOXP2[+]) in *MECP2*-KO ($P = 0.04$; Fig 1H, top) and altered neurotransmitter identity. Transcriptional analysis of genes related to neurotransmission in *MECP2*-KO populations showed changes in glutamatergic, GABAergic, and cholinergic systems ($P < 0.0001$; Fig 1H, bottom, Fig EV3G–K; the full list of cellular markers is shown in Dataset EV1). A summary gene ontology analysis confirmed these findings, showing that alterations in gene expression due to *MECP2*-KO primarily concentrated in neurotransmitter synthesis and receptor pathways in overlapping alignment with the results of our quantitative analysis (Fig 1I).

### Artificial neural network modeling supports targeting synaptic dysfunction for treatment

Although synaptic pathology is a prominent consequence of MeCP2 deficiency (Johnston *et al*, 2003; Gonzales & LaSalle, 2010; Nguyen *et al*, 2012), many compensatory factors may influence network

---

**Figure 1. Human *MECP2*-KO neurons exhibit alterations in synapse-relevant genes and pathways.**

A  Pluripotent stem cells (PSCs) with one of two distinct exonic loss-of-function mutations (Q83X nonsense or K82 frameshift (K82fs)) at the *MECP2* locus were generated and differentiated into neurons via a neural progenitor cell (NPC) intermediate.

B  Immunofluorescent staining confirmed the absence of MeCP2 in *MECP2*-KO PSCs, NPCs, and neurons. Scale bar = 10 μm.

C  RT–qPCR array revealed differential gene expression between control and *MECP2*-KO 28-day neurons. The *P* values are calculated based on a Student's *t*-test of the replicate $2^{-\Delta\Delta Ct}$ values for each gene in the control group and KO groups (WT83/Q83X cell lines were used; $N = 3$ clones from each genotype).

D  Overtime quantitative gene expression profile of many of the involved genes relevant to synaptic function (days 14 and 28 post-differentiation; normalized by control NPC expression).

E  Single-cell RT–qPCR analysis distinguished populations of control and KO 28-day neurons. Principal component analysis (PCA) of 441 cells projected onto the first two components. Overlaid populations of WT83/Q83X and WT82/K82fs neurons are shown.

F  Violin plots of selected genes showing the comparison between control and *MECP2*-KO neurons from the single-cell analyses ($\log_2$(expression) values) that overlapped with the results obtained via RT–PCR array.

G  Volcano plot illustrates differences in expression patterns of target genes of neurons from the qPCR single-cell analyses. The dotted lines represent differentially expressed genes between the groups at $P < 0.05$ (unpaired Student's *t*-test).

H  Left: Schematic of laminar cortical markers. Right: Top: Quantitation of laminar cell markers based on the single-cell RT–qPCR analysis (two-way analysis of variance (ANOVA), $N = 3$ replicates per genotype, $F_{3,24} = 3.31$, *$P = 0.04$, with difference in the proportion of cortical layers V and VI, 95% confidence interval (CI) [8.31, 42.41]); Bottom: altered proportions of neurotransmitter markers (two-way ANOVA, $N = 3$ replicates per genotype, $F_{5,24} = 27.33$, ***$P < 0.0001$, with differences in the proportions of glutamatergic, 95% CI [22.33, 33.60], and cholinergic, 95% CI [0.166, 11.43], neurons).

I  Gene ontology (GO) analysis of targeted genes relevant to synaptic function displaying the differences between *MECP2*-KO and control neurons. For the full list of markers used see Dataset EV1.

Data information: Note that asterisks signify statistically significant difference from *MECP2*-KO. Data are presented as mean ± standard error of the mean (s.e.m.).

---

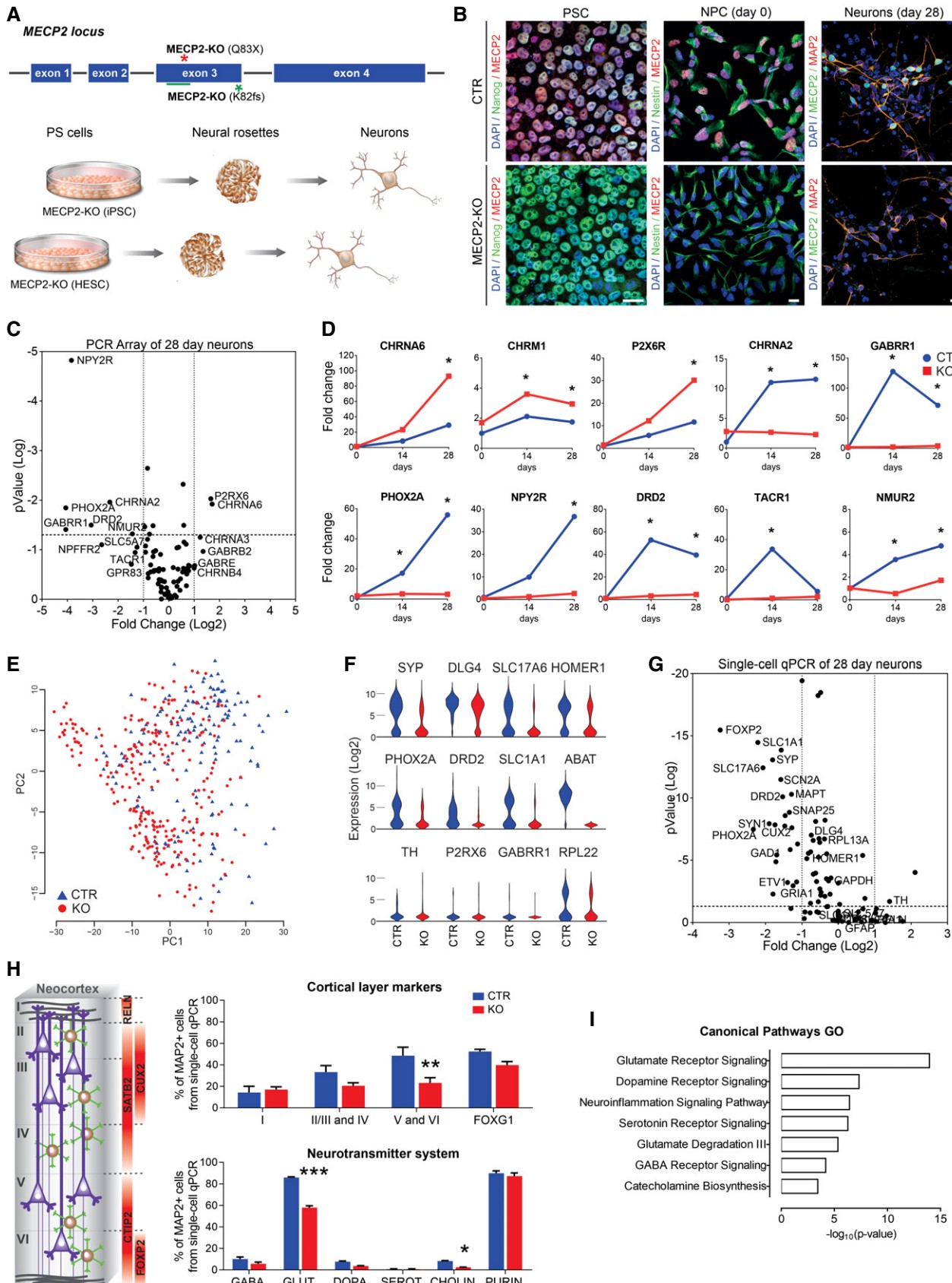

**Figure 1.**

function, and it is unclear that targeted treatment of synaptic dysfunction will yield measurably linked improvement in neuronal population activity. The variation between individuals in network connectivity patterns or the proportions of excitatory and inhibitory neurons, for example, may modulate the link between synaptic phenotype and altered neural activity (Van Vreeswijk & Sompolinsky, 1996; Pena *et al*, 2018). Artificial neural networks offer a biologically plausible framework to explore how parameterized manipulation affects network activity (Miconi, 2017; Kim *et al*, 2019). We generated a neural network *in silico* to predict the change in neuronal spiking activity expected to result from isolated rescue of synaptic structure. We used co-localized synaptic puncta values from a previously published study (Marchetto *et al*, 2010) as a proxy for synaptic knockdown. Considering the full range of possible excitatory-inhibitory ratios (E–I balance) and connection sparsity, we simulated neural activity in networks using synaptic levels commensurate with untreated and rescued *MECP2*-KO. Holding all other parameters equivalent, rescuing the synaptic defect in isolation sufficiently increased neuronal network activity across the parameter space (pairwise delta mean = 0.877, range [−0.002, 7.981]; Fig 2A). The model supports the use of synaptic and neurotransmission phenotypes as actionable targets to improve network function.

**Pharmacological screening identifies two lead compounds that specifically reverse *MECP2*-KO phenotypes**

We selected 14 pharmacological compounds with mechanisms of action that counteract the synapse and neurotransmitter pathologies that we identified in Fig 1 (Fig 2B). For instance, because *MECP2*-KO neurons exhibit cholinergic deficiency (Oginsky *et al*, 2014; Zhang *et al*, 2016; Zhou *et al*, 2017), we included compounds that are predicted to specifically promote this action (e.g., Tacrine,

Carbamoylcholine). We treated *MECP2*-KO and control neurons with each of the 14 compounds and employed a tiered series of assays to identify those that could specifically rescue *MECP2*-KO pathology without affecting controls (Fig 2C). Compounds that either did not treat pathology or significantly altered controls were excluded from use in subsequent assays. In initial assays of synaptogenesis, nine of the 14 compounds increased the quantity of either the presynaptic protein Synapsin 1 or the postsynaptic protein PSD-95, the primary treatment endpoints (Figs 2D and E, and EV4A and B). We evaluated the capacity of these nine compounds to increase the quantity of pre and post co-localized synaptic puncta in *MECP2*-KO neurons, eliminating the drugs that affected control neurons (Figs 2F and G, and EV4C and D).

Our initial screening isolated three drugs (Nefiracetam, Carbamoylcholine, and Acamprosate) that specifically increased synaptogenesis in human *MECP2*-KO neurons (Fig 2D–G). However, the large Carbamoylcholine metabolite has limited gastrointestinal bioavailability and cannot traverse the blood–brain barrier. We elected to instead retain PHA 543613, another direct cholinergic agonist that increased co-localized synaptic puncta with effect nearly identical to that of Carbamoylcholine (PHA 543613, 95% confidence interval (CI) [−8.994, 0.4944] vs. Carbamoylcholine, 95% CI [−9.619, −0.1306]), but which has high oral bioavailability and rapid brain penetration *in vivo* (Wishka *et al*, 2006).

Following the successful increase of protein markers of synaptogenesis and synaptic puncta, we tested the surviving compounds' (Nefiracetam, PHA 543613, and Acamprosate) effects on synaptic function and network activity via calcium imaging and multi-electrode array (MEA) electrophysiology (Nageshappa *et al*, 2016). *MECP2*-KO neurons exhibit less frequent calcium transients that are tractable to manipulators of synaptic function ($P < 0.001$), a decreased percentage of active neurons ($P < 0.001$), and decreased MEA spike frequency (Figs 2H and I and EV4E–G), corroborating

---

**Figure 2.  Screening selected drugs with synaptic action in human neurons identifies Nefiracetam and PHA 543613 as top treatment candidates for MeCP2 deficiency.**

A   *In silico* neural network modeling using our previous synaptic puncta values for untreated (top, knockdown (KD) [6/16] = 0.375*control) and treated (bottom, KD [24/16] = 1.5*control) RTT neurons (Marchetto *et al*, 2010) suggested that isolated increase of synaptic knockdown sufficiently increases neural activity (untreated, mean = 0.665, range = [0.050,1.002]; treated, mean = 1.542, range = [0.997,8.395]). For excitatory-inhibitory (E–I) balance, "0" is fully excitatory, "1" is fully inhibitory.

B   List of compounds for the phenotypic screening. The compounds were selected for mechanisms of action that counteracted the neurotransmitter deficiencies observed in Fig 1. The final concentration was determined based on previous studies.

C   Schematic of drug treatment workflow. Briefly, 28-day-old *MECP2*-KO neurons were treated for 2 more weeks prior to performing phenotypic reversal experiments.

D, E   Western blot quantification showed decreased presynaptic Synapsin1 (D) and postsynaptic PSD-95 (E) in untreated 6-week *MECP2*-KO neurons that can be increased by drug treatment (Kruskal–Wallis test, *$P < 0.05$, **$P < 0.01$, ***$P < 0.001$; full Western blot, Fig EV4A; WT83/Q83X cell lines were used; $N = 3$ clones from each genotype).

F, G   6-week *MECP2*-KO neurons (F) showed a pharmacologically rescuable reduction of co-localized synaptic puncta (G; one-way ANOVA, $F_{16,118} = 9.148$, *$P < 0.05$; Dunnet's multiple comparisons test vs. untreated KO (WT83/Q83X cell lines were used; $N = 7$–8 neurons/condition). *$P < 0.05$, **$P < 0.01$, ***$P < 0.001$. Z scores relative to KO untreated: Control = 3.903; Nefiracetam = 3.560; Carbamoylcholine = 2.705; Pirenzepine = 0.0448; PHA543613 = 2.121; Acamprosate = 3.339; Baclofen = 0.5672; GR73632 = 2.237; Hyperforin = 3.016, and IGF-1 = 3.791. Scale bar = 5 μm.

H   Drug treatment increased calcium transient frequency in 6-week neurons. Fluorescence intensity changes reflecting intracellular calcium transients in neurons in different regions of interest (one-way ANOVA, $F_{4,52} = 20.28$, Dunnett's multiple comparisons test vs. untreated KO: Nefiracetam, **$P < 0.01$ and $Z = 2.364$; PHA 543613, ***$P < 0.001$ and $Z = 3.391$; Acamprosate, ***$P < 0.001$ and $Z = 3.153$; WT83/Q83X cell lines were used; $N = 10$–14 neurons/condition) and the percentage of active neurons (one-way ANOVA, $F_{4,52} = 23.11$; Dunnett's multiple comparison test vs. untreated KO: Nefiracetam, ***$P < 0.001$ and $Z = 3.144$; PHA 543613, **$P < 0.01$ and $Z = 2.567$; **Acamprosate, $P < 0.01$ and $Z = 2.285$; $N = 10$–14 neurons/condition).

I   Treatment with either Nefiracetam or PHA 543613 increased network spiking activity in *MECP2*-KO neurons on MEA (two-sided unpaired Student's *t*-test compared with KO untreated: Nefiracetam, *$P = 0.016$ and $Z = 2.418$; PHA 543613, *$P = 0.027$ and $Z = 2.147$; Acamprosate, $P = 0.39$ and $Z = 0.477$; WT83/Q83X and WT82/K82fs cell lines were used; $N = 4$–11 MEA wells/condition).

Data information: Note that asterisks signify statistically significant difference from *MECP2*-KO untreated. Data are presented as mean ± s.e.m.

---

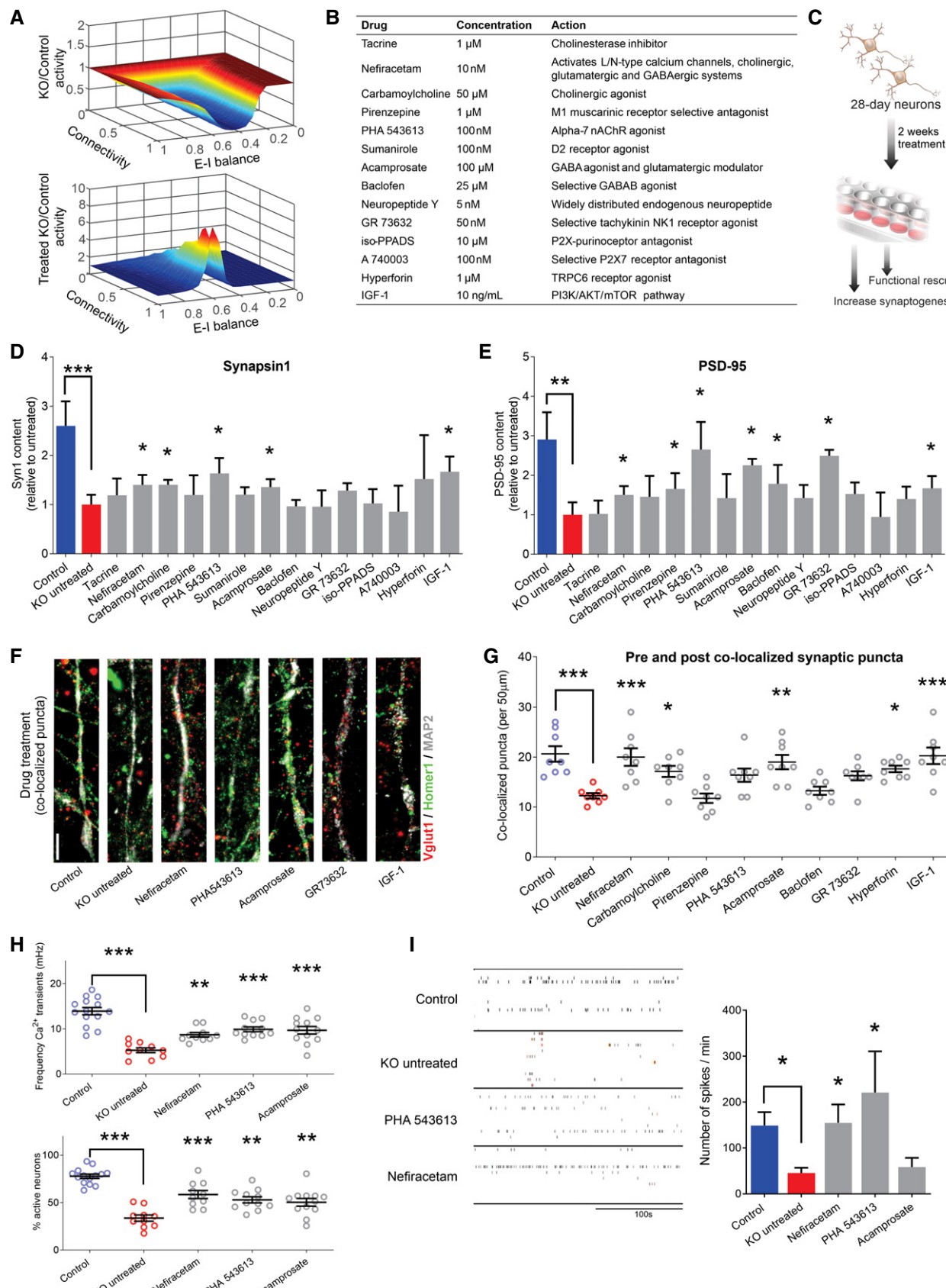

Figure 2.

the predictions of our *in silico* network simulations. Nefiracetam and PHA 543613 significantly increased the frequency of calcium transients and the percentage of neurons that are active (Fig 2H) and restored the MEA spike frequency in *MECP2*-KO neurons (Figs 2 I and EV4E–G). Because Nefiracetam and PHA 543613 specifically rescued synaptic activity in *MECP2*-KO neurons without appreciably affecting controls, these compounds emerged as the most efficacious candidates for further testing in human 3D models.

## Treatment of *MECP2*-mosaic neurospheres increases cell viability and calcium activity

Because *MECP2* resides on the X chromosome, random X-inactivation in females can produce a pattern of cellular genetic mosaicism, with consequent phenotypic gradation, in which some cells express the normal allele and others are mutant (Amir *et al*, 1999; Renthal *et al*, 2018). Therefore, as the next stage to evaluate the two drug candidates, we developed a model of *MECP2* cellular mosaicism (*MECP2*-mosaic) to mimic the female RTT brain by combining control and *MECP2*-KO NPCs in a 50/50 mixture (Fig 3A–C; Sirenko *et al*, 2019). We detected no difference in size (diameter) between untreated *MECP2*-mosaic (CTR/KO) neurospheres and controls (Fig 3D), but both *MECP2*-KO ($P < 0.001$) and *MECP2*-mosaic ($P < 0.001$) neurospheres showed decreased calcium transient amplitude compared with controls (Fig 3E).

Treating *MECP2*-mosaic neurospheres with either Nefiracetam or PHA 543613 did not alter the diameter ($P = 0.79$; Fig 3F). Post-treatment cell viability (representing the inverse of drug-induced cytotoxicity) in the *MECP2*-mosaic neurospheres was improved in a dose-dependent manner by combined treatment with Nefiracetam and PHA 543613 (0.1 μM, 95% CI [−25.94, 0.10], $P > 0.05$; 1 μM, 95% CI [−29.89, −3.85], $P < 0.01$; 10 μM, 95% CI [−39.47, −13.44], $P < 0.001$) (Fig 3G). Calcium transient frequency in *MECP2*-mosaic neurospheres was increased by treatment with Nefiracetam in isolation (1 μM, $P < 0.05$) and when combined with PHA 543613 (10 μM, $P < 0.001$; Fig 3H), but calcium transient amplitude was unaffected by either compound ($P = 0.255$; Fig 3I).

To investigate the impact of the drug treatments on the electrophysiological properties of the *MECP2*-mosaic model, neurospheres treated with 1 μM Nefiracetam, 1 μM PHA 543613, or both Nefiracetam and PHA 543613 were plated to perform MEA analysis. Despite variability in the recordings, we observed an inverse relationship of decreasing spike frequency with an increasing percentage of *MECP2*-KO cells in the neurospheres, but this association did not reach significance ($P = 0.20$). PHA 543613 and Nefiracetam each increased the overall spike count of *MECP2*-mosaic (CTR/KO) neurospheres after 5 weeks of treatment, but because *MECP2*-mosaic neurospheres exhibit similar spiking frequency as control neurospheres ($P = 0.65$), these increases were likewise not significant (PHA 543613 vs. CTR/KO, $P = 0.65$; Nefiracetam vs. CTR/KO, $P = 0.98$; PHA + Nefi vs. CTR/KO, $P = 0.86$; Fig 3J and K). Because each drug conferred a benefit to *MECP2*-mosaic neurospheres, we retained both Nefiracetam and PHA 543613 for further evaluation in a more complex human neurodevelopmental model.

## Nefiracetam and PHA 543613 reverse network activity in *MECP2*-KO cortical organoids

Our final approach to evaluate the therapeutic efficacy of Nefiracetam and PHA 543613 was using cortical organoids, a three-dimensional human neurodevelopmental model that closely recapitulates aspects of human fetal neurodevelopment (Camp *et al*, 2015; Luo *et al*, 2016; Trujillo *et al*, 2019). We generated *MECP2*-KO and control cortical organoids as previously described (Figs 4A and EV5A and B; Trujillo *et al*, 2019). One-month-old cortical organoids were treated for another month with either Nefiracetam (1 μM) or PHA 543613 (1 μM), and experiments and analyses were performed using organoids at 2–3 months of age. Both Nefiracetam ($P < 0.001$) and PHA 543613 ($P < 0.001$) increased the diameter of *MECP2*-KO organoids (Fig 4B), which is reduced despite their equivalent areas of Ki67$^+$ proliferation ($P = 0.73$; Fig 4C).

RNA sequencing and gene ontology analyses in two-month-old organoids revealed that both Nefiracetam and PHA 543613 promoted expression of genes involved in pathways relevant to

**Figure 3.   Treatment of *MECP2*-mosaic neurospheres with Nefiracetam and PHA 543613.**

A       Schematic showing the formation of control, *MECP2*-KO, and mosaic *MECP2*-KO/control neurospheres (WT83/Q83X cell lines were used).

B, C    Immunofluorescent staining (B) and quantitation (C) confirmed the graded decrease in MeCP2 expression from control to 50% *MECP2*-mosaic to full *MECP2*-KO neurospheres. Scale bar = 50 μm.

D       8-week *MECP2*-KO neurospheres ($N = 44$) exhibited decreased size (diameter; **$P < 0.01$), but 50% *MECP2*-mosaic neurospheres (CTR/KO; $N = 44$) were unchanged from controls (one-way ANOVA, $F_{2,150} = 6.03$, $P = 0.003$; Dunnett's multiple comparisons test; $N = 65$). Left: representative bright-field images of neurospheres; Right: neurosphere size quantification. Scale bar = 200 μm.

E       Calcium imaging analysis showing calcium transients normalized by control (one-way ANOVA, $F_{2,72} = 15.62$, $P < 0.0001$; Dunnett's multiple comparisons test vs. control: *MECP2*-mosaic, ***$P < 0.001$; *MECP2*-KO, ***$P < 0.001$). WT83/Q83X cell lines were used; $N = 44$ Q83X neurospheres and 65 WT83 neurospheres.

F       8-week neurosphere size (diameter) remained unchanged by treatment with either drug (one-way ANOVA, $F_{6,344} = 0.52$, $P = 0.79$; $N = 44$ Q83X neurospheres and 65 WT83 neurospheres).

G       Cell viability was improved or unharmed in treated 8-week neurospheres (one-way ANOVA, $F_{9,104} = 6.81$; Dunnett's multiple comparisons test vs. CTR/KO untreated: Nefi + PHA 1 μM, **$P < 0.01$ and $Z = 2.974$; Nefi + PHA 10 μM, ***$P < 0.001$ and $Z = 3.733$; $N = 8$–24 neurospheres per condition).

H, I    Calcium transient frequency (number of peaks in 10 mins recording) normalized by CTR/KO untreated (H; one-way ANOVA, $F_{9,239} = 6.18$; Dunnett's multiple comparisons test vs. CTR/KO untreated: Nefi 1 μM, *$P < 0.05$ and $Z = 0.780$; Nefi + PHA 10 μM, ***$P < 0.001$ and $Z = 1.972$; $N = 15$-61 neurospheres per condition) and calcium transient amplitude normalized by CTR/KO untreated (I; one-way ANOVA, $F_{9,241} = 1.27$, $P = 0.255$; $N = 16$–62 neurospheres per condition).

J, K    MEA spike frequency heatmap (J) and quantification (K) normalized by CTR/KO untreated (one-way ANOVA, $F_{5,38} = 1.21$, $P = 0.321$; Dunnett's multiple comparisons test vs. CTR/KO untreated: PHA, $P = 0.65$ and $Z = 1.134$; Nefi, $P = 0.98$ and $Z = 0.06$; PHA + Nefi, $P = 0.86$ and $Z = 0.94$; $N = 7$-8 wells per condition). 'X' on activity heatmap signifies absence of a neurosphere in that position.

Data information: Nefi: Nefiracetam; PHA: PHA 543613. Note that asterisks in F–K signify a statistically significant difference in treated *MECP2*-mosaic (CTR/KO) neurospheres compared with untreated CTR/KO neurospheres. Data are presented as mean ± s.e.m.

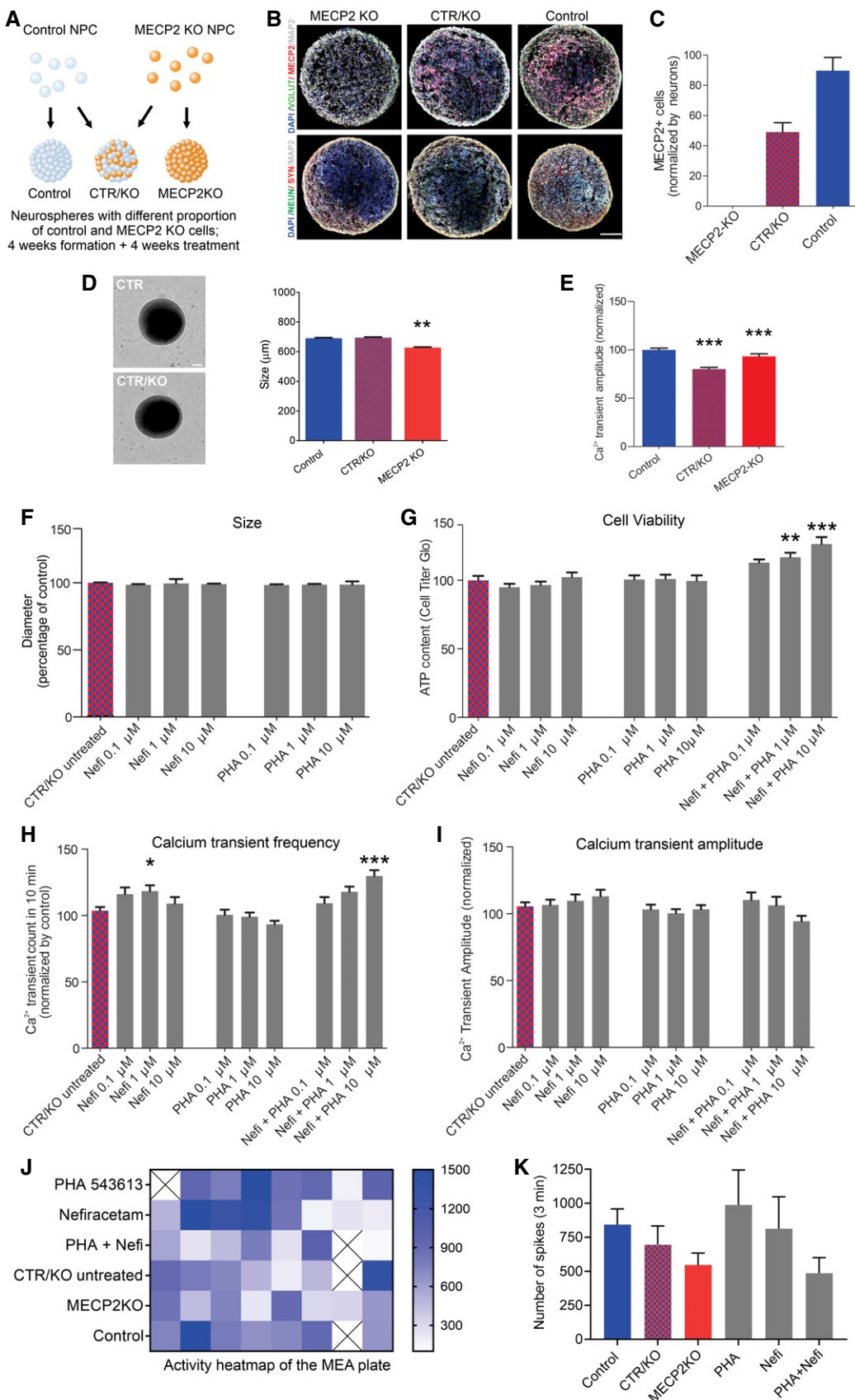

Figure 3.

synaptic function (Figs 4D–G and EV5C and D). The compounds were able to increase gene expression of specific neurotransmitter markers including cholinergic, GABAergic, and glutamatergic signaling. These findings were supported by the capacity of Nefiracetam ($P < 0.01$), albeit not PHA 543613, to increase synaptic puncta (Fig 4H and EV5E and F). In 2- to 3-month-old *MECP2*-KO and control organoids, an equivalent percentage of cells immunostained positive for the neuronal marker NeuN ($P = 0.80$; Fig 4I), and the cortical layer V and VI marker CTIP2 was unaffected by drug treatment ($P = 0.07$; Fig 4J). Although the primary aim of the drug-screening pipeline was to increase synaptogenesis and activity in *MECP2*-KO neurons, morphological features were also investigated. Nefiracetam and PHA 543613 did not increase the number or length of neurites and did not rescue nuclei size. However, we observed an increase in neuronal spine-like protrusions due to the treatments (Fig EV5G).

To further evaluate network activity, we used cortical organoids to perform MEA electrophysiology, as we previously demonstrated (Trujillo *et al*, 2019). *MECP2*-KO organoids exhibited decreased population spiking compared with the controls ($P < 0.01$), but treating the *MECP2*-KO organoids with Nefiracetam and PHA 543613 each increased population spiking to a level not significantly different from that of control organoids (Nefiracetam, $P = 0.98$ in comparison to control organoids; PHA 543613, $P = 0.12$ in comparison with control organoids; Fig 4K–M).

## Discussion

In the present study, we used human *MECP2*-KO PSCs to undertake the goals of identifying gene-specific pathology and therapeutically targeting pathways downstream from *MECP2* loss-of-function (Braat

& Kooy, 2015; Pozzo-Miller *et al*, 2015; Katz *et al*, 2016; Benke & Möhler, 2018). Our analyses of *MECP2*-KO neurons revealed altered expression of synapse-relevant genes, findings that corroborate the contribution of synaptic impairment to RTT pathology (Chao *et al*, 2007; Dani & Nelson, 2009; Chao *et al*, 2010; Krishnan *et al*, 2017; Chen *et al*, 2018; Banerjee *et al*, 2019). After identifying targetable neuropathology, supported by computational simulations of network behavior, we developed a human-based drug-screening platform comprised of increasingly complex models that closely mimic human fetal neurodevelopment, thereby facilitating clinical translatability. We strategically employed a tiered series of assays and models (monolayer neurons, *MECP2*-mosaic neurospheres, and cortical organoids) to screen 14 targeted pharmacological compounds. Two of these, Nefiracetam and PHA 543613, specifically treated *MECP2*-KO pathophysiology while leaving controls unaffected (Fig 5).

Our *MECP2*-KO neuronal cultures showed that synaptic and neurotransmitter pathophysiology principally concentrated in glutamatergic and cholinergic dysregulation. Clinically, the lead compounds Nefiracetam and PHA 543613, both of which are orally administered, have invaluable mechanisms of action to treat these deficiencies. Nefiracetam is a cholinergic, GABAergic, and glutamatergic agonist developed to enhance cognitive functioning (Malykh & Sadaie, 2010; Moriguchi, 2011), and PHA 543613 is an α7-nAChR agonist with proven neuroprotective effects in a neurodevelopmental disease model (Foucault-Fruchard *et al*, 2018). Cholinergic modulatory effects within the nervous system are many because nAChRs are widely dispersed across the neuronal and synaptic architecture, and acetylcholine additionally affects the release of other neurotransmitters (Picciotto *et al*, 2012). Although pronounced GABAergic pathology has been observed in mice (Chao *et al*, 2010), we observed lesser GABAergic contribution in our

---

**Figure 4. Treatment increases synapse-related gene expression and network activity in *MECP2*-KO cortical organoids.**

A Schematic of PSC aggregation and development into cortical organoids.

B 2-month-old cortical organoid diameter (one-way ANOVA, $F_{2,173} = 91.07$, $P < 0.0001$; Dunnett's multiple comparisons test vs. untreated: Nefiracetam, ***$P < 0.001$ and $Z = 5.414$; PHA 543613, ***$P < 0.001$ and $Z = 5.351$; WT83/Q83X and WT82/K82fs cell lines were used; $N = 28$–90 organoids per condition). Left: representative bright-field images of cortical organoids; Right: cortical organoid size quantification. Scale bar = 200 μm.

C Ki67 + proliferative area (Student's *t*-test, $t_{10} = 0.36$, $P = 0.73$, $N = 6$ each). Left: representative immunofluorescence images of progenitor, proliferative, and neuronal markers in cortical organoids; Right: quantification of Ki67 + proliferation in cortical organoids. Scale bar = 50 μm.

D, E RNA sequencing (D, left: heatmap) and GO analysis (D, right) of untreated *MECP2*-KO organoids vs. Nefiracetam ($N = 4$ samples each of ~10–15 pooled organoids) indicated that treatment upregulated genetic expression in synapse-relevant pathways. E, center band of the box is median log$_2$ normalized expression and edges are 25th and 75th percentiles; whiskers are minimum and maximum log$_2$ normalized expression values.

F, G RNA sequencing (F, left: heatmap) and GO analysis (F, right) of untreated *MECP2*-KO organoids vs. PHA 543613 ($N = 4$ samples each of ~10–15 pooled organoids) indicated that treatment upregulated expression of genes in synapse-relevant pathways. G, center band of the box is median log$_2$ normalized expression and edges are 25th and 75th percentiles; whiskers are minimum and maximum log$_2$ normalized expression values.

H Synapsin1 + puncta quantitation (one-way ANOVA, $F_{3,34} = 10.07$, $P < 0.0001$; Dunnett's multiple comparisons test vs. untreated: Nefiracetam, **$P < 0.01$ and $Z = 2.998$; PHA 543613, $P > 0.05$ and $Z = 1.266$; WT83/Q83X cell lines were used; $N = 8$–10 each). Left: representative image of Synapsin1 + immunofluorescent staining; Right: Synapsin1 + puncta quantification. Scale bar = 20 μm.

I, J *MECP2*-KO and control organoids had similar proportions of cells positive for the neuronal marker NeuN (I; Student's *t*-test, $t_{14} = 0.26$, $P = 0.80$, $N = 8$ each) and cortical layer V and VI neuronal marker CTIP2 (J; one-way ANOVA, $F_{3,40} = 2.489$, $P = 0.074$ and $Z < 1$, WT83/Q83X and WT82/K82fs cell lines were used; $N = 8$–14 organoids each). J, left: representative image of CTIP2 + immunofluorescent staining in cortical organoids; J, right: quantification of CTIP2 + neurons. Scale bar = 50 μm.

K–M *MECP2*-KO cortical organoids plated on MEA (K; left: schematic depicts process of plating organoids on MEA; right: top-down view of organoid on MEA; scale bar = 200 μm) showed decreased population spiking (L; left and center: representative population spiking traces of control and *MECP2*-KO organoids; right: quantification of organoid population spiking activity, $N = 6$ replicates each). Drug treatment increased spiking network activity of *MECP2*-KO cortical organoids to a level not significantly different from that of control organoids (M; one-way ANOVA, $F_{3,31} = 3.775$, Dunnett's multiple comparisons test vs. control: untreated KO, **$P < 0.01$ and $Z = 3.192$; Nefiracetam, $P = 0.98$ and $Z = 2.117$; PHA 543613, $P = 0.12$ and $Z = 1.101$; WT83/Q83X cell lines were used; $N = 7$–10 MEA wells per condition).

Data information: Note that asterisks signify a statistically significant difference from untreated *MECP2*-KO cortical organoids. Data are presented as mean ± s.e.m.

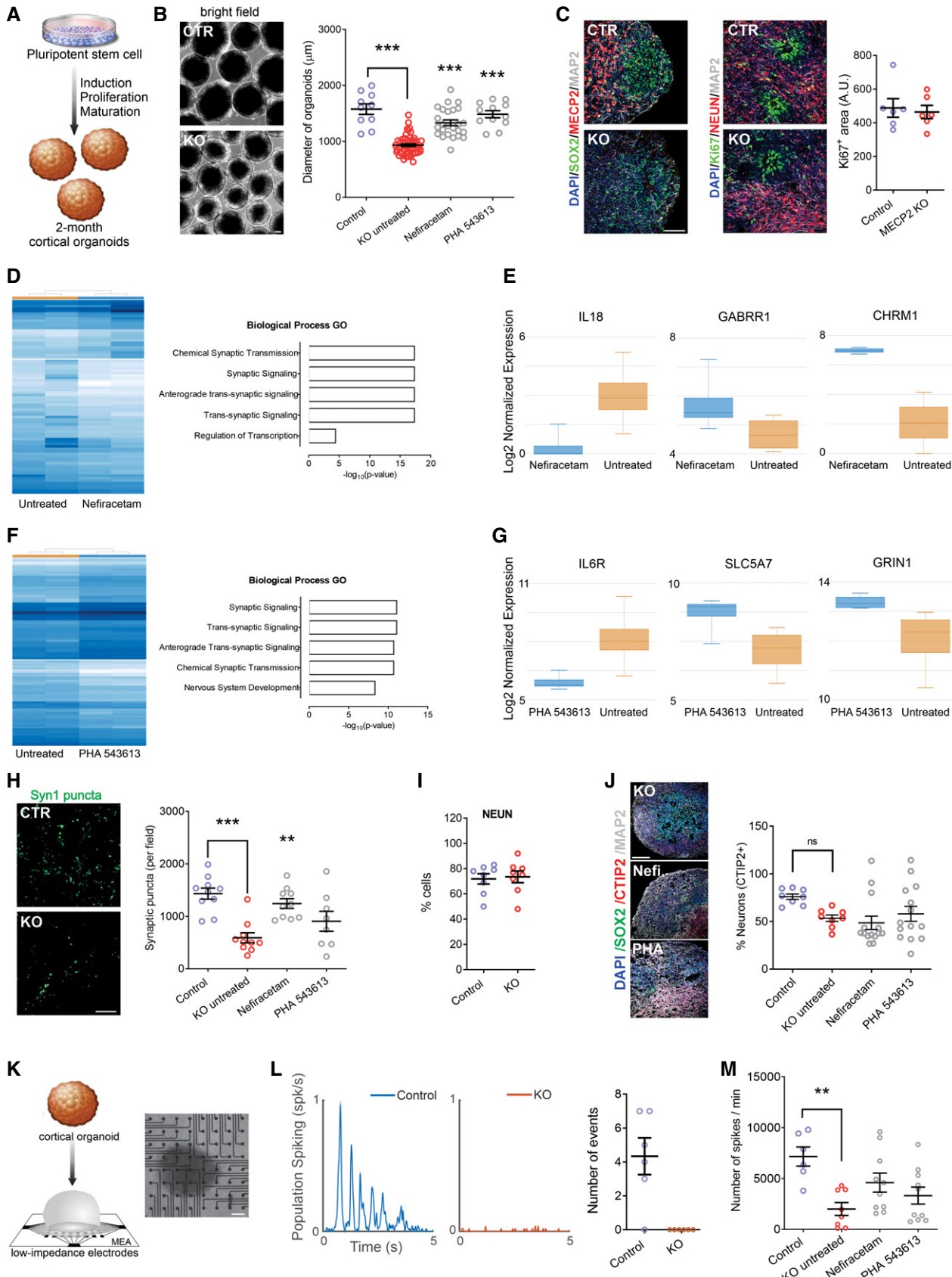

**Figure 4.**

human-based neuronal cultures, and compounds that mainly modulate GABA failed our drug-screening pipeline. We do not observe high proportions of GABAergic neurons in our 3D systems (organoids and neurospheres), so the compounds' effects that we observed here are presumably via their cholinergic and, for Nefiracetam, glutamatergic actions. Nevertheless, inhibitory dysregulation likely contributes to *MECP2*-KO pathophysiology, and future studies of RTT that employ human-based cell models with high GABAergic representation could explore this aspect of the pathology.

Several completed and ongoing clinical trials have explored therapeutic safety or efficacy in RTT patients, mainly of IGF-1 or modulators of BDNF, the encoding gene of which is a target of MeCP2 (Chen *et al*, 2003). Although formulations of IGF-1 are expected to be promising candidates (Pozzo-Miller *et al*, 2015), results thus far have been mixed. Recombinant human IGF-1 (mescarmin) was safe and mildly efficacious in a phase 1 trial of RTT patients (clinicaltrials.gov NCT01253317; Khwaja *et al*, 2014), but a double-blind, placebo-controlled follow-up study of mescarmin did not show significant phenotypic improvements (clinicaltrials.gov identifier NCT01777542; O'Leary *et al*, 2018). In contrast, the trial of a synthetic form of IGF-1 (trofinetide; NCT02715115; Glaze *et al*, 2017) demonstrated sufficient safety and efficacy to warrant initiation of a phase 3 trial (NCT04279314). Meanwhile, clinical trials that have preliminarily investigated modulators of BDNF have shown efficacy of glatiramer acetate (NCT02153723; Djukic *et al*, 2016) and led to further trial of fingolimod (NCT02061137). Of the two compounds we identified in the present study, Nefiracetam has previously entered clinical trials for other disorders involving cholinergic deficiency, including dementia, post-stroke apathy, and major depressive disorder (Fujimaki *et al*, 1993; Robinson *et al*, 2009). Thus, the ready availability of Nefiracetam and PHA 543613 and their capacity to specifically improve the neurocytopathology identified in our human *MECP2*-KO models encourages suggestion of clinical efficacy and therapeutic trial for patients with neurodevelopmental disorders.

Past research has investigated the capacity of pharmacological compounds to rescue RTT pathophysiology using *in vivo* mouse models (Tang *et al*, 2019). Several recent studies highlight that despite general conservation, homologous human and mouse cell types exhibit extensive differences, including alterations in proportion, distribution, gene expression, and neurotransmission. Notably, the most different systems are glutamatergic, serotoninergic, and cholinergic (Hodge *et al*, 2019; Sjostedt *et al*, 2020). These species-specific features emphasize the lack of clinical translatability and the importance of directly studying human brain models. In addition, as was alluded to above, the safety of Nefiracetam and PHA 543613 have already been evaluated in animal models, and the safety of Nefiracetam has been demonstrated in human populations, being commercially available for human use (Fujimaki *et al*, 1993; Robinson *et al*, 2009; Foucault-Fruchard *et al*, 2018). We directly demonstrated the effects of these compounds in a human context with a variety of *MECP2*-KO human cell models. Our laboratory has shown that the most sophisticated of these models, cortical organoids, develop oscillatory activity similar to that observed during human fetal neurodevelopment (Trujillo *et al*, 2019). Due to the synaptic and network impairment resulting from MeCP2 deficiency, development of similar oscillatory activity likely occurs much later in *MECP2*-KO organoids, but documentation and characterization of such activity in future studies may reveal itself to be a useful biomarker in a clinical setting.

Previous research in human RTT neurons associated MeCP2 deficiency with a reduction in total RNA (Li *et al*, 2013), suggesting that increased synaptic expression following treatment may be secondary to increased total RNA. However, our transcriptional analyses showed reduced expression of genes concentrated in pathways relevant to synaptic function, and our process of screening synaptically targeted compounds retained only those that increased these deficient pathways in KO cells without affecting controls, suggesting that increased expression of synaptic genes is a primary effect of the targeted therapeutics. Regardless, MeCP2 deficiency causes a spectrum of severity of synaptic phenotypes, both between and within individuals, and it is this fact that encouraged our development of a cellular model of *MECP2* mosaicism. Our innovative human-based neural model of cellular mosaicism has clear utility for *MECP2* and other X-linked genetic disorders, and the concept can likewise be expanded to include other aspects of pharmacogenomic precision medicine, such as characterizing pharmacological response or variation in cytotoxicity for patients with novel mutations. Furthermore, the impact of somatic mosaicism on clinical and neurodevelopmental phenotypes is becoming increasingly appreciated, so this human model will offer further utility for the study of somatic neuronal mosaicism in other neuropsychiatric diseases and human neurodevelopment in general (Lodato *et al*, 2015; McConnell *et al*, 2017).

Our study has several intrinsic limitations. One limitation is that, as a proof-of-concept study for our drug-screening platform, we tested only a restricted number of compounds. Because we selected candidate drugs such that their mechanisms of action were directed against the pathological alterations we had identified, any compounds not explicitly mentioning a desired action would have

**Figure 5. Drug-screening pipeline overview.**

Top: Left: *MECP2*-KO neurons exhibit aberrant synaptic transcriptomics and synaptic morphology as well as decreased calcium transients and neural network activities. Right: Pharmacological compounds were selected for screening based on the expected ability of their mechanisms of action to counteract the synaptic and neurotransmitter pathologies we identified. Center: With step-wise progression through a pipeline of assays (shown as dark-to-light shades of blue), drugs were eliminated either if they did not reverse the *MECP2*-KO pathology identified in the assay or if they affected controls. Nefiracetam and PHA 543613 emerged as the two compounds that could reverse *MECP2*-KO neurocytopathology without affecting controls. Middle: The effects of Nefiracetam and PHA 543613 were validated in two 3D human cell models, *MECP2*-mosaic neurospheres (Left) and *MECP2*-KO cortical organoids (Right). Left: One or both drugs reversed calcium transient frequency and amplitude as well as MEA spike frequency in *MECP2*-mosaic neurospheres, but not their size. Right: One or both drugs reversed organoid size, SYN1 puncta, and MEA spike frequency in *MECP2*-KO cortical organoids. Bottom: A schematic proposing a hypothetical mechanism by which Nefiracetam and PHA 543613 might influence synaptic function.

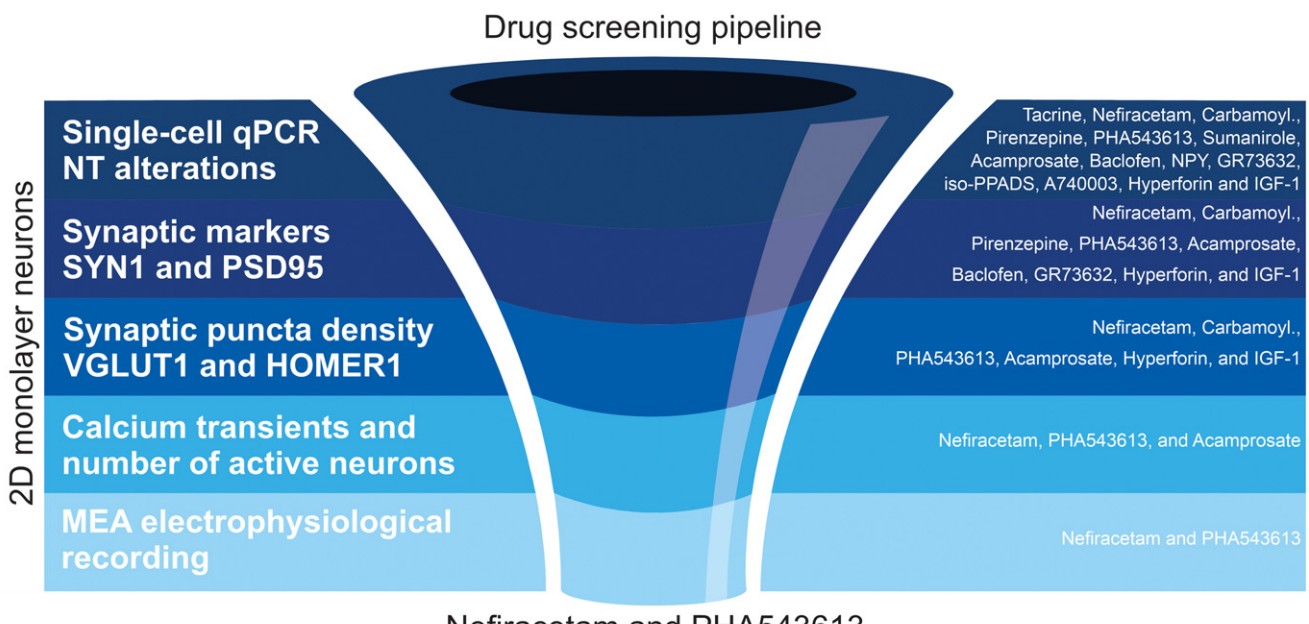

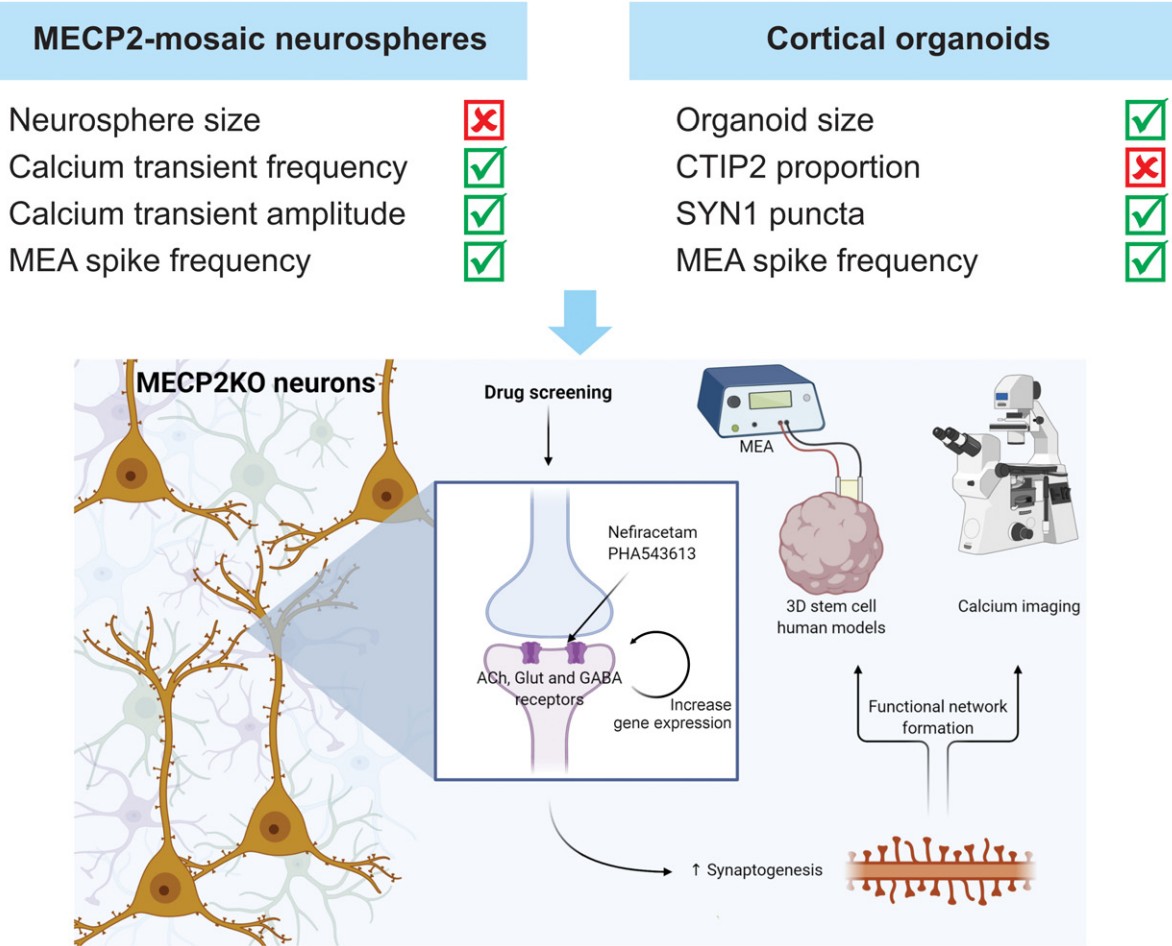

**Figure 5.**

gone unconsidered. A future study employing a platform with similar complexity as ours but with high-throughput screening capability may identify additional efficacious compounds. A second limitation is that the drugs were administered during an early neurodevelopmental period, even in the cortical organoids (Camp *et al*, 2015; Luo *et al*, 2016; Trujillo *et al*, 2019), which may be challenging to assess in a clinical setting. Trial of these compounds in an expanded age range may help optimize their clinical utilization, aided by the fact that the control models were not significantly affected by treatment with these drugs.

As would be expected in any pharmacological context, the selected compounds were unable to fully reverse all MeCP2-deficient phenotypes. Although Nefiracetam and PHA 543613 increased the MEA network activity of *MECP2*-KO cortical organoids, they did not fully restore activity, suggesting other deficiencies also contribute to *MECP2*-KO network pathology, synaptic and otherwise, that must be investigated and targeted with other compounds. In addition, neither compound significantly increased calcium transient amplitude in *MECP2*-mosaic neurospheres, but the compounds did increase their calcium transient frequency, a finding that aligns with our observation that Nefiracetam and PHA 543613 increased calcium transient frequency and activity in *MECP2*-KO neurons. The clinical implication of calcium frequency vs. amplitude in *MECP2*-mosaic neurospheres is unclear and presents an important direction for future study, but the concordance of our findings between assays and models affirms our confidence in the compounds' effects. Neither compound, moreover, increased the proportion of CTIP2 + cells, a not unexpected result considering the drugs were selected to counteract synaptic pathology, not neuronal fate. It is unclear why populations of *MECP2*-KO monolayer neurons contained a smaller proportion of layer V/VI neurons. However, previous research has noted that these neurons are profoundly heterogeneous in their morphology, gene expression, axonal projection, and electrophysiological function and that *MECP2* may affect any of these parameters in a context- and region-specific manner (Stuss *et al*, 2012). It would be reasonable to expect that a smaller proportion of these neurons would reduce synaptic maturation and connectivity, but correcting the mechanism responsible for this deficit requires compounds whose function is not restricted to the synapse and may require intervention at an earlier timepoint. Treatment of diseases incident to MeCP2 deficiency may thus be improved with a combination of drugs, another reason to explore a more comprehensive array of compounds.

Despite the limitations, we believe the findings of our study warrant a pilot trial of Nefiracetam and/or PHA 543613 in a clinical setting of RTT or another disorder of MeCP2 deficiency. Our drug-screening platform was developed to facilitate clinical translatability. We used a series of increasingly complex human models with the final, cortical organoids, closely recapitulating *in vivo* human fetal neurodevelopment, and we eliminated drugs that significantly affected controls to limit either clinical error or adverse clinical effects. Nefiracetam is already available for human use commercially, and it has undergone extensive safety testing in human populations (Fujimaki *et al*, 1993). Neurodevelopmental disorders caused by MeCP2 deficiency are severe and treatment options are limited or altogether unavailable, so our findings that Nefiracetam and PHA 543613 improve synaptic morphology, genetic expression, and network activity suggest that, for some

patients, these pharmacological compounds may offer a meaningful clinical impact.

# Materials and Methods

### Generation of *MECP2*-KO cell lines and cell culture

Control PSCs were generated and characterized as previously described (Gore *et al*, 2011; Nageshappa *et al*, 2016; Trujillo *et al*, 2019). Of the two MeCP2-deficient PSC lines, one was generated via pluripotent induction of male patient-derived fibroblasts as described (Marchetto *et al*, 2010), and the other was created via CRISPR/Cas9 introduction of a frameshift mutation into the *MECP2* locus of H9 hESCs (Thomas *et al*, 2017), creating an early stop codon and non-functional MeCP2. We used exome sequencing to confirm mutagenesis and analyze off-target effects. PSC colonies were maintained on Matrigel-coated dishes (BD-Biosciences, San Jose, CA, USA) and fed daily with mTeSR1 (Stemcell Technologies, Vancouver, Canada) as we have previously described (Chailangkarn *et al*, 2016); only mycoplasma-negative cell cultures were used. Cells were evaluated by karyotype and CNV arrays to detect genomic alteration. Informed consent was obtained from all human subjects, and experiments conformed to the principles set forth in the WMA Declaration of Helsinki and the Department of Health and Human Services Belmont Report. The study was approved by the UCSD IRB/ESCRO committee (protocol 141223ZF).

### Karyotyping

Molecular Diagnostics Service (San Diego, CA, USA) performed G-banding karyotyping analyses.

### Teratoma assay

PSC colonies were dissociated, centrifuged, resuspended in 1:1 Matrigel:PBS, and subcutaneously injected into nude mice. Teratomas were excised, fixed, and sliced after 1-2 months, and sections were stained with H&E. Protocols were previously approved by the UCSD Institutional Animal Care and Use Committee.

### Neuronal differentiation of PSCs

Differentiation was performed as previously described (Chailangkarn *et al*, 2016). After seven days, mTeSR1 used to feed PSCs was replaced with N2 medium [DMEM/F12 (Life Technologies, Carlsbad, CA, USA), 0.5× N2 (Life Technologies), 1% penicillin/streptomycin (P/S; Life Technologies), 1 μM Dorsomorphin (Dorso; R&D Systems, Minneapolis, MN, USA), and 10 μM SB431542 (SB; Stemgent, Cambridge, MA, USA)] for 1–2 days, after which embryoid bodies (EBs) were formed by scraping PSC colonies and culturing them in shaker suspension (95 rpm at 37°C) for eight days. EBs were dissociated and plated on a Matrigel-coated dish in N2B27 medium [N2 medium with 0.5× B27-Supplement (Life Technologies) and 20 ng/ml FGF-2]. Emerging rosettes were picked manually, dissociated in Accutase (Life Technologies), and seeded on poly-L-ornithine/laminin plates. Emergent NPCs were expanded and

maintained in N2B27 medium with feeding on alternate days; all NPCs used for neurons were passage 5–20. FGF-2 was withdrawn from the medium to induce neuronal differentiation, considered Day 0 of differentiation.

## Calcium imaging in monolayer neurons

Calcium imaging was performed as described (Chailangkarn *et al*, 2016). Briefly, PSC-derived neuronal networks were transduced a lentivirus Syn::RFP reporter construct. Cultures were washed with Krebs HEPES buffer and incubated with Fluo-4AM (Molecular Probes/Invitrogen, Carlsbad, CA, USA). A Hamamatsu ORCA-ER digital camera (Hamamatsu Photonics, Japan) with 488 nm filter on an Olympus IX81 inverted fluorescence confocal microscope (Olympus Optical, Tokyo, Japan) acquired 5000 frames at 28 Hz in a $256 \times 256$-pixel region (100× magnification). Images were acquired with MetaMorph7.7 (MDS Analytical Technologies, Sunnyvale, CA, USA) and analyzed using ImageJ and Matlab7.2 (Mathworks, Natick, MA, USA) using *PeakCaller* script. A 95th percentile threshold of amplitude for calcium spikes was set for event detection; signal amplitude is presented as fluorescence change ($\Delta F/F$) following background subtraction.

## *MECP2*-mosaic neurospheres

StemoniX, Inc. provided human *MECP2*-mosaic cortical neural spheroids as customized microBrain®3D Assay Ready 384-Plates (Sirenko *et al*, 2019). Briefly, each well of a 384-well plate contained a single, free-floating human PSC-derived neurosphere comprised of a balanced culture of cortical neurons and astrocytes generated from a 50/50 mixture of NPCs obtained from a male patient with Rett syndrome (Q83X mutation in the *MECP2* gene) and its sex-matched parental control. Four-week-old neurospheres were treated with Nefiracetam (0.1, 1, or 10 µM), PHA 543613 (0.1, 1, or 10 µM), or a combination of both, for an additional 4 weeks. Bright-field images of the neurospheres were taken using an Image Xpress Micro Confocal High-Content Imaging System (Molecular Devices LLC, San Jose, CA, USA) and the size measured using an automated plug-in in the MetaMorph Microscopy Automation and Imaging Analysis Software (Molecular Devices LLC).

## Calcium transients assay in neurospheres

Intracellular $Ca^{2+}$ transients of *MECP2*-KO, *MECP2*-mosaic, and control neurospheres were measured using a fluorescence-based assay, as previously described (Sirenko *et al*, 2019). Briefly, eight weeks after differentiation, neurospheres were incubated with FLIPR Calcium-6 dye for two hours at 37°C, 5% $CO_2$. Next, the cultures were transferred to a FLIPR Tetra High-Throughput Cellular Screening System (Molecular Devices LLC) to determine the kinetics of intracellular $Ca^{2+}$ transients. With the instrument at 37°C, the recording was performed for 10 min at a frequency of 2 Hz, exposure time per read of 0.4 s, excitation intensity of 40%, and with the gain set to 30. The analysis was performed using ScreenWorks Peak Pro (4.2) software package (Molecular Devices LLC) and included the quantitative evaluation of transient peak count and average transient peak amplitude (Sirenko *et al*, 2019). A 95th percentile threshold of amplitude for calcium spikes was set for event detection; signal amplitude is presented as fluorescence change ($\Delta F/F$) following background subtraction.

## Cell viability assay in neurospheres

The viability of neurospheres after eight weeks of differentiation, untreated and treated with Nefiracetam and/or PHA 543613, was established using the CellTiter-Glo 3D Viability Assay (Promega, Madison, WI, USA), according to the manufacturer instructions. Briefly, a CellTiter-Glo 3D Reagent volume equal to the volume of media present in each well was added to the neurospheres. After mixing vigorously for five minutes to promote cell lysis, the plate was incubated for 25 additional minutes at room temperature. The luminescence signal was recorded using a PHERAstar FSX plate reader (BMG Labtech Inc., Cary, NC, USA).

## Generation of cortical organoids

Cortical organoids were generated as previously described (Trujillo *et al*, 2019). Briefly, PSCs cultured for approximately seven days were dissociated with 1:1 Accutase (Life Technologies):PBS, and cells were plated into a 6-well plate ($4 \times 10^6$ cells/well) in mTeSR1 supplemented with 10 µM SB (Stemgent), 1 µM Dorso (R&D Systems), and 5 µM Y-27632 (EMD-Millipore, Burlington, MA, USA) and cultured hereafter in shaker suspension (95 rpm at 37°C). Formed spheres were fed mTeSR1 (with 10 µM SB and 1 µM Dorso) for three days followed by Media1 [Neurobasal (Life Technologies), 1× Glutamax (Life Technologies), 2% Gem21-NeuroPlex (Gemini Bio-Products, Sacramento, CA, USA), 1% N2-NeuroPlex (Gemini Bio-Products), 1% non-essential amino acids (NEAA; Life Technologies), 1% P/S (Life Technologies), 10 µM SB, and 1 µM Dorso] for six days, every other day; Media2 (Neurobasal, 1× Glutamax, 2% Gem21, 1% NEAA, and 1% P/S) with 20 ng/ml FGF-2 (Life Technologies) for seven days, daily; Media2 with 20 ng/ml each of FGF-2 and EGF (PeproTech, Rocky Hill, NJ, USA) for 6 days, every other day; and Media2 with 10 ng/ml each of BDNF, GDNF, and NT-3 (all PeproTech), 200 µM L-ascorbic acid (Sigma-Aldrich, St. Louis, MO, USA), and 1 mM dibutyryl-cAMP (Sigma-Aldrich) for 6 days, every other day. Cortical organoids were subsequently maintained indefinitely in Media2 without supplementation.

## Pharmacological treatment

One-month-old neurons were treated for two weeks at the drug concentrations listed in Fig 2B. Neurospheres were treated as detailed above. One-month-old cortical organoids were treated for one month with either Nefiracetam (1 µM) or PHA 543613 (1 µM) during every-other-day media changes. The concentration of the drugs was selected based on the specific $K_d$ reported for each drug and from previous work on animal models and human neurons. The time points were strategically selected after we observed a cellular or functional phenotype in *MECP2*-KO neurons. Based on published work from our group, *MECP2*-KO neurons and neurospheres already show a distinct deficit after 4 weeks.

## RNA extraction and RT–qPCR array

RNA was extracted using an RNeasy Mini Kit (Qiagen, Hilden, Germany) according to the manufacturer's instructions. DNase-treated RNA was assessed with a NanoDrop 1000 (Thermo

Scientific, Waltham, MA, USA). Total RNA was converted to cDNA and evaluated for RT–PCR array per the manufacturer's instructions (Qiagen, Hilden, Germany). The full list of markers used can be found in Dataset EV1.

### Single-cell qRT–PCR

Single-Cell and BioMark HD Systems (Fluidigm, San Francisco, CA, USA) were used for specific target amplification in NPCs, neurons, and organoids as described (Chailangkarn et al, 2016; Trujillo et al, 2019). Briefly, single cells were captured on a C1 chip, and a LIVE/DEAD Cell Viability/Cytotoxicity kit (Life Technologies) was used to assess viability. DELTAgene primer pairs (96.96 Dynamic Array IFC chip) were used for single-cell qPCR; results analysis was performed using Fluidigm Real-time PCR Analysis Software and Singular Analysis Toolset 3.0. The full list of markers used can be found in Dataset EV1.

### RNA sequencing analyses

An RNeasy Mini kit (Qiagen) was used to isolate RNA for library preparation (Illumina TruSeq RNA Sample Preparation Kit; San Diego, CA, USA) and sequencing (Illumina HiSeq2000, 50bp paired-end reads, 50 million high-quality sequencing fragments per sample, on average). Data were analyzed by Rosalind (https://rosalind.onramp.bio/), with a HyperScale architecture (OnRamp BioInformatics, Inc., San Diego, CA). Reads were trimmed using cutadapt, quality scores were assessed using FastQC, and reads were aligned to the Homo sapiens genome build hg19 using STAR. Individual sample reads were quantified using HTseq and normalized via Relative Log Expression using DESeq2 R library. DEseq2 was also used to calculate fold changes and p values and perform optional covariate correction. Gene clustering of differentially expressed genes was done using the Partitioning Around Medoids method using the fpc R library. Functional enrichment analysis of pathways, gene ontology, domain structure, and other ontologies was performed using several databases (HOMER, Interpro, NCBI, MSigDB, REACTOME, WikiPathways).

### Gene ontology (GO)

RNA sequencing enrichment was performed with the WebGestalt (Zhang et al, 2005) and Cytoscape (Shannon et al, 2003) plug-ins utilizing statistically significant categories ($P < 0.05$). Genes evaluated for differential expression set the background for GO annotation and enrichment analysis.

### Immunofluorescence of cells and cortical organoids

Immunofluorescence was performed as previously described (Chailangkarn et al, 2016; Trujillo et al, 2019). Cells were fixed in 4% paraformaldehyde, washed three times with PBS (5 min each), and permeabilized/blocked (0.1% Triton X-100 and 2% BSA in PBS). Cortical organoids were fixed overnight in 4% paraformaldehyde and then transferred to 30% sucrose, sunken, embedded in O.C.T. (Sakura, Tokyo, Japan), and cryostat-sectioned at 20 μm. Slides with organoid sections were air-dried and then permeabilized/blocked (0.1% Triton X-100 and 3% FBS in PBS). Primary antibodies (goat

anti-Nanog, Abcam (Cambridge, UK) ab77095, 1:500; rabbit anti-Lin28, Abcam ab46020, 1:500; mouse anti-MeCP2, Sigma-Aldrich M7443, 1:500; rabbit anti-Oct4, Abcam ab19857, 1:500; mouse anti-Nestin, Abcam ab22035, 1:200 (organoid: 1:250); rat anti-CTIP2, Abcam ab18465, 1:250 (organoid: 1:500); rabbit anti-SATB2, Abcam ab34735, 1:200; chicken anti-MAP2, Abcam ab5392, 1:1,000 (organoid: 1:2,000); rabbit anti-Synapsin1, EMD-Millipore AB1543P, 1:500; mouse anti-Vglut1, Synaptic Systems (Goettingen, Germany) 135311, 1:500; rabbit anti-Homer1, Synaptic Systems 160003, 1:500; mouse anti-NeuN, EMD-Millipore MAB377, 1:500; rabbit anti-Ki67, Abcam ab15580, 1:1,000; rabbit anti-SOX2, Cell Signaling Technology (Danvers, MA, USA) 2748, 1:500; rabbit anti-GFAP, DAKO (Carpinteria, CA, USA) Z033429, 1:1,000; rabbit anti-TBR1, Abcam ab31940, 1:500; rabbit anti-TBR2, Abcam ab23345, 1:500; rabbit anti-beta-catenin, Abcam E247, 1:200; mouse anti-GABA, Abcam ab86186, 1:200; rabbit anti-PROX1, Abcam ab101651, 1:250) in blocking buffer incubated overnight at 4°C. Slides were washed three times with PBS (5 min each) and incubated with secondary antibodies (Alexa Fluor 488, 555, and 647, Life Technologies, 1:1,000 in blocking buffer). Nuclei were stained with DAPI (1:10,000). Slides were mounted with ProLong Gold anti-fade mountant (Life Technologies) and analyzed with a Z1 Axio Observer Apotome fluorescence microscope (Zeiss, Oberkochen, Germany).

### Artificial neural network simulations

Simulations of neural network activity were done with artificial recurrent neural networks using biologically plausible parameter choices, defined by five free parameters: number of neurons in the network ($N$), connection sparsity or percent connectivity ($p_{con}$), percentage of inhibitory vs. excitatory neurons ($p_{inh}$), synaptic weights ($W$), and synaptic decay time constants of the neurons ($\tau$). A network size of $N = 512$ neurons was used; simulations performed with larger and smaller network sizes yielded qualitatively similar results. Percent connectivity ($p_{con}$) refers to the percentage of all possible pairwise connections in the network for which a connection was present (non-zero entry in $W$). Our networks followed Dale's Law, which states that any given neuron is either excitatory or inhibitory. For each simulation, an excitatory-inhibitory balance was chosen determining the percent of inhibitory neurons ($p_{inh}$).

Our artificial neural network implementation was similar to the continuous rate networks outlined in Kim et al (Kim et al, 2019). Briefly, each simulated network consisted of a pool of neurons that could be connected to any other neuron in the pool. Each neuron received weighted input from other neurons and integrated these inputs to produce a firing rate. The network dynamics are governed by the equation:

$$\tau \frac{dx}{dt} = -x + Wr \qquad (1)$$

where $\tau \in \mathbb{R}^{1 \times N}$ is the synaptic decay time constant for the $N$ neurons in the network, $x \in \mathbb{R}^{1 \times N}$ is the synaptic current variable, $W \in \mathbb{R}^{N \times N}$ is the matrix of synaptic weights between all $N$ neurons, and $r \in \mathbb{R}^{1 \times N}$ is the output firing rates. The output firing rate is given by an element-wise nonlinear transformation of the synaptic current variable. We used the standard logistic sigmoid function per Kim et al (Kim et al, 2019):

$$r = \frac{1}{1 + \exp(-x)} \qquad (2)$$

The synaptic connectivity matrix $W$ was randomly initialized from a normal distribution with mean of zero and standard deviation of $g/\sqrt{N \cdot p_{con}}$, where $g$ is the gain term. We set $g = 15$ because previous studies have shown that networks operating in a high gain regime ($g \geq 1.5$) support rich dynamics analogous to those of biological networks (Laje & Buonomano, 2013; Kim *et al*, 2019). Synaptic decay time constants were randomly initialized to a value in the biologically plausible range of 20–100 ms. As in Kim *et al* (Kim *et al*, 2019), we used the first-order Euler approximation method to discretize equation (1) for the simulations.

To vary synaptic knockdown or strengthening, we introduced a synaptic weight scaling factor, $p_{KD}$. Based on Marchetto *et al* (2010), synaptic strength was decreased in a PSC model of RTT, particularly for excitatory neurons, so we perturbed the synaptic strength of excitatory synaptic connections only. To model synaptic knockdown or strengthening, we multiplied the excitatory synaptic weights by $p_{KD}$. Each network simulation was generated with randomly initialized synaptic weights and synaptic time constants, as described above, and then run for 200 time steps (one time step simulates 5 ms). For each parameter combination, $[p_{con}, p_{inh}, p_{KD}]$, 100 independent simulations were run, and the average of all neurons' firing rates on the last time step was taken from each simulation as a measure of network activity. We then took the average network activity across all 100 simulations as an estimate for the network activity for the given set of parameters. For every $[p_{con}, p_{inh}]$ pair, we ran simulations for the $p_{KD}$ values of interest as well as a "control" network for which $p_{KD} = 1$ (i.e., no synaptic scaling was applied). All measures of network activity were normalized to that of the corresponding control network to obtain the relative network activity.

To create the relative activity surface plots, we ran the simulations described above for all networks with $p_{con} \in \{1e-5, 0.05, 0.1, 0.15, \ldots, 0.95, 1.0\}$ and $p_{inh} \in \{0.0, 0.05, 0.1, \ldots, 0.95, 1.0\}$, representing 441 $[p_{con}, p_{inh}]$ pairs tiling the possible space of network connectivity and excitatory-inhibitory balance. Simulations across this space were run for each of the synaptic perturbation conditions taken from Marchetto *et al* (2010): RTT-like synaptic knockdown ($p_{KD} = 0.375$), IGF-1-like synaptic improvement ($p_{KD} = 1.5$), and control ($p_{KD} = 1$). A minimum value of $p_{con} = 1e-5$ was used because $p_{con}$ must be greater than zero to calculate the standard deviation of the random initialization distribution for the synaptic weights, as described above.

## Western blotting

Cells were lysed in RIPA buffer with protease inhibitor, and total protein was extracted and quantified (Pierce BCA Protein Assay Kit, Thermo Scientific). Total protein (20μg) was separated using a Bolt 4–12% Bis-Tris Plus Gel (Life Technologies) and transferred to a nitrocellulose membrane using an iBlot2 dry blotting system (Thermo Scientific). Membranes were blocked for 1–4 h (Rockland Immunochemicals, VWR International, Arlington Heights, IL, USA); primary antibodies (rabbit anti-Synapsin1, EMD-Millipore AB1543P, 1:1,500; mouse anti-PSD-95, Neuromab, 1:1,500; chicken anti-MAP2, Abcam ab5392, 1:2,000; rabbit anti-GFAP, DAKO Z033429, 1:2,000) in blocking buffer incubated, shaking, overnight at 4°C and secondary antibodies (IRDye 680RD and IRDye 800CW, 1:5,000 in blocking buffer) for 1 h at room temperature. Proteins were detected by an Odyssey CLx infrared imaging system (LiCOR Biosciences, Lincoln, NE, USA), and semi-quantitative analysis of signal intensity was corrected to the relative quantification of β-actin.

## Synaptic puncta quantification

Co-localized Vglut1 (presynaptic) and Homer1 (postsynaptic) immunostained puncta along MAP2-positive processes were quantified as described (Nageshappa *et al*, 2016). Primary antibodies incubated 2 h, secondary antibodies incubated 1h, and coverslips were mounted; see above for complete immunofluorescence methodology. The slides were imaged using a Z1 Axio Observer Apotome fluorescence microscope (Zeiss), and a blinded investigator manually quantified co-localized synaptic puncta along 50 μm segments of randomly selected MAP2-positive processes (7–8 per condition); for quantification, no distinction was made between proximal and distal segments.

## Analysis of spine density, stability, and motility

pHIV7/Syn-EGFP lentivirus was prepared by transfecting HEK cells using the transfection agent polyethylenimine (Polysciences, Inc., Warrington, PA, USA), Syn-EGFP, and packaging plasmids pMDL, Rev RSV, and VSVG. Control and *MECP2*-KO neurons were transduced with pHIV7/Syn-EGFP on day 40 of differentiation and allowed to further differentiate until day 70, after which they were fixed with 4% paraformaldehyde, and processed for imaging. Confocal images were obtained using a confocal laser-scanning microscope (Nikon, 60× oil) and sequential acquisition setting at a resolution of 4,096 × 4,096 pixels. Each image is a Z-series projection of ~7 to 15 images, averaged twice and taken at 0.5 μm depth intervals. GFP-labeled differentiated neurons were chosen randomly for quantification. Dendrites on primary branches were selected randomly, and two to three segments of 10 μm were analyzed for each neuron. Morphometric measurements were performed manually using the ImageJ software, and dendritic spines were manually traced by a blinded investigator. Nuclei size was manually measured using ImageJ.

## Analysis of neuronal morphology

PSC-derived neuronal morphometry was performed as previously described (Chailangkarn *et al*, 2016). Neurons with branching neurites featuring dendritic spine-like protrusions were randomly selected, and neuronal morphology was quantified using Neurolucida v9 (MBF Bioscience, Williston, VT) via Nikon Eclipse E600 microscope (40× oil). Measurements were as follows: spine number: number of spines protruding from neurites; soma size: soma area on cross-section; and spine length: summed length of all spines per neuron. The rater traced the same neuron after to ensure intra-rater reliability.

## Multi-electrode array analysis

Multi-electrode array (MEA) analyses of neurons and cortical organoids were performed as described (Nageshappa *et al*, 2016; Trujillo *et al*, 2019). Briefly, neurospheres were plated on dual-chamber MEA; spontaneous spike activity was evaluated with the MED64

**The paper explained**

**Problem**

The X-linked gene *MECP2* encodes an epigenetic regulatory protein that is critical for typical human brain development. *MECP2* deficiency causes severe neurodevelopmental impairment that clinically presents most commonly as Rett syndrome. However, despite pinpointing the genetic disruption and the eminent treatability of the disease, no clinically approved treatments for Rett syndrome are currently available.

**Results**

By establishing an innovative drug-screening pipeline using human pluripotent stem cells differentiated into neurons, *MECP2*-mosaic neurospheres, and cortical organoids, we isolated two lead compounds that specifically reversed synaptic and network pathology resulting from *MECP2* deficiency while leaving controls unaffected. The two compounds, Nefiracetam and PHA 543613, improved synaptic morphology and function in *MECP2*-KO human neurons, increased calcium activity in *MECP2*-mosaic neurospheres, and reversed synaptic transcriptomic pathways and neural network function in *MECP2*-KO cortical organoids.

**Impact**

*MECP2* deficiency causes profound neurodevelopmental impairment that urgently demands treatment. This study identified two pharmacological lead compounds, one of which has already been approved for human use commercially, that could specifically reverse *MECP2*-KO neurocytopathology in human cell models. These compounds may offer meaningful clinical impact for some patients with severe neurodevelopmental impairment resulting from *MECP2* deficiency, and trial in a clinical setting is warranted following regulatory approval.

System (Panasonic). Glutamatergic (AP5, NBQX), GABAergic (Gabazine), and gap channel (Mefloquine) antagonism were used to verify neuronal activity. Recorded spikes were analyzed using the Neuroexplorer software (Nex Technologies, Madison, AL, USA). For cortical organoids, one-month-old organoids were plated on MEA plates and recordings were collected with a Maestro MEA system and AxIS Software Spontaneous Neural Configuration (Axion Biosystems, Atlanta, GA, USA). The plate was incubated in the machine for three minutes before five minutes of recording. Axion Biosystems's Neural Metrics Tool classified as "active" those electrodes with at least five spikes/minute. Bursts were identified using an inter-spike interval (ISI) threshold requiring a 5-spike minimum and 100ms maximum ISI. Network bursts required a minimum of 10 spikes under the same ISI and at least 25% active electrodes.

**Statistical analysis**

Statistical analyses were performed using GraphPad Prism (GraphPad Software, La Jolla, CA), except the *in silico* neural network model, the procedure for which is detailed above. Sample sizes were determined based on previous publications from this laboratory and others. Experiments were performed in at least three independent replicates and different cell lines; experiment-specific information is detailed in the figure legends. Samples were allocated and evaluated according to genotype; no randomization was applied. Analyses of synaptic puncta and dendritic spine tracing were made blinded. Data exclusion in MEA datasets (outliers) was carried out automatically using pre-established criteria as described above. Outliers in other experiments were determined using GraphPad criteria and excluded. Results for

continuous variables were expressed as mean $\pm$ standard error of the mean and 95% confidence intervals were normal-based. Normality was assessed visually, and variance was accounted for in all analyses. Means for continuous variables were compared between groups using, where appropriate, unpaired Student's *t*-test, one-way, or two-way analyses of variance, and nonparametric distributions were compared using Kruskal–Wallis test. Tests were performed two-sided with $\alpha$ throughout set as 0.05.

# Data availability

The datasets and computer code produced in this study are available in the following databases:

- RNA-seq data for cortical organoids: Gene Expression Omnibus GSE160146 (https://www.ncbi.nlm.nih.gov/geo/query/acc.cgi?acc=GSE160146)
- Network modeling computer scripts: GitHub (https://github.com/tsudacode/in_silico_bioANN)

Expanded View for this article is available online.

## Acknowledgments

The authors gratefully acknowledge Dr. Kristen Jepsen at the UCSD Institute of Genomic Medicine, funded from the National Institutes of Health (NIH) grant (#S10 OD026929). This work was supported by grants from the California Institute for Regenerative Medicine TR2-01814 and TR4-06747; the NIH through P01 NICHD033113, NIH Director's New Innovator Award Program 1-DP2-OD006495-01, R01MH094753, R01MH103134, U19MH107367, 5T32GM007198; a CARTA Fellowship to J.W.A.; and a NARSAD Independent Investigator Grant to A.R.M. C.A.T. was partly funded by NIAAA K01AA026911 and a Loulou Foundation grant. HU acknowledges grant support by the São Paulo Research Foundation (FAPESP 2018/07366-4), Brazil.

## Author contributions

Experiment design and manuscript writing: CAT and JWA; C1 single-cell analyses, calcium imaging, spine density, cell number, proliferation, and apoptosis: CAT; Synaptic quantification: CAT, LT, AA, and ChAT.; Statistical analyses: CAT and JWA; Cortical organoid generation and characterization, RNA extraction and analyses of sequencing and gene ontology, and the MEA recordings: JWA, CAT, and PDN; MEA data analysis: CAT and PDN.; Ingenuity Pathways Analysis and Western blots: PDN.; *in silico* network model design: BT and TJS; Simulations and analysis: BT; Cell viability and calcium activity development, characterization, and analysis in the mosaic neurospheres: PDN, CC, NS, KMF, SR, and FZ; Study conceptualization: HU and ARM; Manuscript review: All authors.

## Conflict of interest

Dr. Muotri is a co-founder and has equity interest in TISMOO, a company dedicated to genetic analysis and brain organoid modeling focusing on therapeutic applications customized for autism spectrum disorder and other neurological disorders with genetic origins. The terms of this arrangement have been reviewed and approved by the University of California San Diego in accordance with its conflict of interest policies.

## For more information

- *MECP2* OMIM entry: https://www.omim.org/entry/300005.
- USA National Institutes of Health information on Rett syndrome: https://

www.ninds.nih.gov/Disorders/Patient-Caregiver-Education/Fact-Sheets/Rett-Syndrome-Fact-Sheet.
- Nefiracetam PubChem reference: https://pubchem.ncbi.nlm.nih.gov/compound/Nefiracetam.
- PHA 543613 PubChem reference: https://pubchem.ncbi.nlm.nih.gov/compound/56972222.
- USA Food and Drug Administration (FDA) Clinical Trials Guidance Documents: https://www.fda.gov/regulatory-information/search-fda-guidance-documents/clinical-trials-guidance-documents.

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
