## [Review Process File · EMBO Molecular Medicine]

Pharmacological reversal of synaptic and network pathology in human MECP2-KO neurons and cortical organoids

Cleber Trujillo, Jason Adams, Priscilla Negraes, Cassiano Carromeu, Leon Tejwani, Allan Acab, Ben Tsuda, Charles Thomas, Neha Sodhi, Katherine Fichter, Sarah Romero, Fabian Zanella, Terrence Sejnowski, Henning Ulrich, and Alysson Muotri

DOI: [10.15252/emmm.202012523](https://doi.org/10.15252/emmm.202012523)

Corresponding authors: [Alysson Muotri \(muotri@ucsd.edu\)](mailto:muotri@ucsd.edu) , [Cleber Trujillo \(ctrujillo@ucsd.edu\)](mailto:ctrujillo@ucsd.edu)

Review Timeline:

Submission Date:	14th Apr 20
Editorial Decision:	25th May 20
Revision Received:	31st Aug 20
Editorial Decision:	30th Sep 20
Revision Received:	26th Oct 20
Accepted:	28th Oct 20

Editor: *Zeljko Durdevic*

Transaction Report:

25th May 2020

Dear Dr. Muotri,

Thank you for the submission of your manuscript to EMBO Molecular Medicine. We have now heard back from the three referees who agreed to evaluate your manuscript. As you will see from the reports below, the referees acknowledge the interest of the study. However, they raise some concerns that should be addressed in a major revision of the present manuscript. Particular attention should be given to better characterization of the effects of Nefiracetam and PHA 543613 in vitro and in organoids and to validation of the findings in an in vivo model. Addressing the reviewers' concerns in full will be necessary for further considering the manuscript in our journal.

Acceptance of the manuscript will entail a second round of review. Please note that EMBO Molecular Medicine encourages a single round of revision only and therefore, acceptance or rejection of the manuscript will depend on the completeness of your responses included in the next, final version of the manuscript. For this reason, and to save you from any frustrations in the end, I would strongly advise against returning an incomplete revision.

We realize that the current situation is exceptional on the account of the COVID-19/SARS-CoV-2 pandemic. Therefore, please let us know if you need more than three months to revise the manuscript.

I look forward to receiving your revised manuscript.

***** Reviewer's comments *****

Referee #1 (Remarks for Author):

Trujillo et al. employs human induced pluripotent and embryonic stem cell-derived neurons, neurospheres and cortical organoids models to test the effects of two lead compounds, Nefiracetam and PHA 543613, on rescuing the synaptic defects of Rett syndrome (RTT) caused by mutations of X-linked MECP2 gene. Although the usage of human stem cell-derived technologies is highly relevant and represents a suitable model for drug screening and selection, there are flaws in the experimental design. The authors show that the two lead compounds can enhance the expression of synaptic markers, yet the effects on network activities are not convincing. In the

Discussion, the authors pointed out three intrinsic limitations of this study that were poorly resolved. It is unclear how the two lead compounds may help relieve symptoms of Rett syndrome, and thus limits the clinical relevance of this study. Since the mouse model of Rett Syndrome is widely available from Jackson lab, the authors should test the rescue effects of the two lead compounds in *Mecp2* KO and/or Het mice, at least for several hallmark defects such as synaptic markers level and spine morphology/density.

Major comments:

-Nefiracetam and PHA 543613 increase expression of synaptic markers Synapsin 1 and PSD-95. Many papers show that deficits of RTT are not only at the level of synaptic proteins. A series of papers show that GABA dysregulation is a major contributing factor in RTT, at least in mice (Chao et al., 2010; Banerjee et al., 2016, Durand et al., 2012). According to existing literatures, GABAergic deficits seem to be more important than glutamatergic neurotransmission. Can the two drug leads proposed by the authors affect inhibition?

-It provides very limited information when the authors only use synaptic protein level as a major measurement of rescue effects. Other major cellular hallmarks of RTT should be examined. Supple Fig2 describes defects in spine-like morphology such as reduced spine number, spine motility and spine stability. Can the drug leads rescue this defect?

-The authors use different readouts to assess network effects in different in vitro models, which make it very difficult to assess the effects. Figure 2h,i measures calcium oscillation frequency in stem cell-derived neurons, while Figure 3f and g measures calcium peak frequency and peak amplitude. The definition of calcium peak frequency is unclear and should be described in more detail. Figure 4 i-k uses MEA to measure population spiking in organoids and the rescue effect in Fig 4k is NOT significant. The reviewer suggests the authors to use standardized measurements to assess the individual or network neuronal activities. That is crucial to validate the effects of their two compound leads.

-Reduced total RNA has been suggested to be a major cellular deficit in ESC-derived RTT neurons (Li et al., Cell Stem Cell, 2013). Do the two drug leads rescue total RNA level? Could the increase in synaptic markers be a secondary effect of total RNA level rescue?

-Cortical organoid model is a good model, but the effect of Nefiracetam and PHA 543613 should also be tested in vivo using *Mecp2* knockout mice to demonstrate the effect of the two drug leads at the physiological level. The knockout mouse model of Rett Syndrome is widely available from Jackson lab. The authors should test the rescue effects of the two lead compounds in *Mecp2* KO and/or Het mice, at least for several hallmark defects such as synaptic markers level and spine morphology/density.

-In Fig 1a and supple Fig 1, the authors show that both iPSC and HESC were used to generate two knockout models (Q83X and K82fs) for the production of neuroprogenitors, neurons and cortical organoids. However, the particular line used in later figures and legends were not made clear; the authors merely used *MECP2*-KO lines. This is not acceptable since iPSC and HESC are different stem cell models. The authors should specify the line used in each figure for the scientific community to assess the effects of the drug and whether there is line-specific results.

Other comments:

-It is not useful to compare the size of neurosphere (supple Fig 5) and organoids (Fig 4b) since the

size of neurospheres and organoids can vary within and between batches generated at different times. Data on similar organoid size is only shown for wild type (supple Fig 6b). The authors could include a similar graph on knockouts. The authors should compare individual soma size in their 2D monolayer, neurosphere and organoids models, like supple Fig 2d. and in the author's earlier paper (Marchetto et al., 2010).

-The authors should include more description on how the 14 drug leads were initially selected. Although there is currently no available treatment for RTT, the author should also include a brief discussion on drug leads that are under clinical trials such as IGF-1 and cite the relevant papers.

-On page 7, the authors mention that MECP2-KO neurons exhibit cholinergic deficiency to justify the use of Nefiracetam and PHA 543613. The authors should cite relevant papers.

-The simulation in Fig 2a is interesting, but it does not provide too much information that is clinically relevant to the symptoms of RTT. It is reasonable to predict that rescuing synaptic defects would enhance neuronal network activities. This piece of information aligns with published experimental findings, but is limited in novelty.

-More description of Supple Fig 6f and g should be included in the legend and in text. The meaning of the figure is not clear.

-Methodology can be in more details. Correct grammatical errors.

- Since the manuscript contains only 4 main figures. It may be better suited to be reformatted as a short report with 3 figures.

Referee #2 (Remarks for Author):

Trujillo and colleagues investigated the cellular, synaptic, and functional properties of human stem cell-derived neurons, neurospheres, and organoids produced from control and isogenic CRISPR/Cas9-engineered lines with two different MeCP2 mutations. They confirmed synaptic and other deficits in mutant cells and identified promising pharmacological treatments to compensate those deficits. Specifically, they showed that MECP2-KO neurons show morphological abnormalities in soma area and spine density, as well as alterations in synaptic gene expression and calcium dynamics. Based on the gene expression data, the authors selected 14 candidate compounds for therapeutic intervention, and then the two most promising ones for further testing. In MECP2-mosaic neurospheres, treatments with either drug did not reduce cell viability, but did improve calcium frequency but not amplitude. In MECP2-KO organoids that showed decreased size as compared to control organoids, treatments with Nefiracetam or PHA543613 treatment. Based on the findings in this manuscript, the authors draw the conclusion that a pilot trial of Nefiracetam and/or PHA 543613 in treating MECP2 deficiency is warranted. Overall, the manuscript is strong and the claims are supported by the data. I have only few minor-to-moderate suggestions for improvement:

1. The authors should specify the names and origin of control stem cell lines used in the experiments.

2. The authors should specify throughout the manuscripts which lines and from how many

differentiations were used in different experiments, and at which developmental timepoints those experiments were performed. For example, in Fig. 1, it is unclear which lines were used for PCR and immunostaining assays and when those assays were performed (Fig. 1g-h). In addition, the authors should describe why those primers were selected. It is surprising that only few excitatory synaptic markers were tested in those experiments. Accordingly, the claim "Many of the implicated genes appeared relevant to synaptic function, and the differences in expression became increasingly pronounced as differentiation progressed" (page 5), does not seem to be supported by the data and should be revised.

3. It is unclear what experiments were performed to characterize cell-type specific marker expression in Fig 1h. I assume it is immunostaining; however, it is not stated neither in the text nor in the figure legend. If it is immunostaining, the authors should present some representative images supporting the quantifications. In addition, the authors should discuss why reduced proportions of layer V and VI cortical neurons were detected in MeCP2-KO and how this result might affect synaptic maturation and connectivity.

4. The significance of GO analysis in Fig. 1 is unclear. Only few selected genes were tested in PCR experiments, and the panel of excitatory synaptic genes seems to be incomplete.

5. Authors should provide more details on the differentiation protocols used in this study. For example, it is unclear for how many days neural progenitors were treated with FGF2.

6. The rationale and value of network modeling is unclear. It is expected that increased synaptic transmission should lead to increased probability of neuronal spiking. Can this model predict the extent to which synaptic transmission should be increased to improve spiking but not to induce seizures?

7. The authors used synaptic staining as a primary assay to detect and rescue synaptic deficits. It is important to know the robustness of this assay. The authors should calculate and report Z-factor (Zhang, Chung, and Oldenburg J Biomol Screen. 1999).

8. The authors should provide more details on the synaptic assay (Fig. 2). Which dendrites (proximal or distal) were selected for analysis? How many dendrites were analysed. Has this been done manually or automatically by a blinded or non-blinded investigator?

9. The authors should specify when different experiments on organoids were performed. For example, they report that ~80% of cells in organoids expressed NeuN. NeuN is a mature neuronal marker. When was this measured?

10. Untreated organoids in Fig 4d and Fig. 4e show drastically different gene expression profiles. Why is that? The authors should specifically discuss the similarity and differences in expression networks altered by Nefiracetam and PHA543613 treatments. In addition, the authors should present gene expression data obtained in isogenic control organoids. Otherwise, it is unclear whether the treatments rescue gene expression profiles.

11. The authors should provide high-magnification images of synapses in Fig. 4f. The presented images are of poor quality. They should also discuss how synapses were quantified in organoid sections. Where and how the ROIs were selected.

12. In the previous study, the authors investigated network oscillations in stem cell-derived

organoids. It would be to show the comparison of oscillations in control and MeCP2-deficient organoids. This might serve as a biomarker in the future clinical studies.

13. The authors should discuss why Nef and PHA treatment do not rescue network deficits in organoids (Fig 4k shows p value > 0.05).

14. The sample size should be stated for all quantifications including those in Fig. 1.

15. Scale bars are not defined or missing on most of the images.

16. Figure S3b: Text is unreadable.

Referee #3 (Remarks for Author):

In the present work, the authors take advantage of previously observed phenotypes associated with MECP2-KO neurons, such as synaptic formation and neuronal maturation, to assess pharmacological compounds able to (partially) reverse those phenotypes. The authors performed an extensive array of assessments of neurons and organoids, from genomic to morphological and functional approaches. One very elegant aspect of the present study was the use of both 2D and 3D human stem cell models. In total, there were 14 drugs selected based upon their mechanism of action, targeting different synaptic proteins. From those, Nefiracetam and PHA 543613, specifically partially rescued synaptogenesis defects, calcium signaling, and impaired neural network activity. The manuscript is well written and organized and the data is presented in a clear fashion. However, it is important to highlight that, while the overall story and subject are relevant, there are concerns regarding MECP2-mosaic neurospheres and the selection of the final drugs that should be addressed.

Major Concerns:

1- The authors initially selected 14 compounds based on their mechanisms of action associated with synapsis and neurotransmitters. However, the mechanism connecting the drugs, especially Nefiracetam and PHA 543613, to MECP2 was not investigated in this work. That information would be important to assess if the efficiency of the drugs are specific to MECP2 KO pathogenesis and should be better explored in the discussion.

2- A very elegant addition to the work of the MECP2-mosaic model, mimicking the female RTT syndrome brain. However, no rationale was given as to why the authors chose to mix NPCs and work with neurospheres. A more relevant model would be the cortical organoids which could be achieved by mixing the iPSCs. In the same context, calcium influx was used as a readout for the MECP2-mosaic model. Calcium activity was used as a proxy for the neurotransmitters defects, however, it would be crucial to assess if their electrophysiology potential is different and if that could also be rescued using Nefiracetam and PHA 543613 or the combination of those drugs.

3- The majority of the experiments that assessed the efficacy of the drugs only displayed a partial rescue, including the electrophysiology of the cortical organoids, therefore, the text should be edited to properly represent that. I.e. Introduction: "Two currently available lead compounds, Nefiracetam and PHA 543613, exhibited therapeutic potential to rescue the synaptic and functional network defects caused by MeCP2 deficiency and are viable candidates for clinical trial" (page 4) should read "...exhibited therapeutic potential to partially rescue the synaptic and functional network.."

Minor Concerns

- 1- It was not clear in the results or methods' sections how were the concentrations of the drugs selected. Was there a concentration curve for these drugs other than in figure 3? The same question could be applied for the selected time points presented in this work.
- 2- In the result section entitled: "Treatment of MECP2-mosaic neurospheres increases cell viability and calcium activity" the authors describe the MECP2-mosaic as a combination of "different proportions of control and MECP2-KO NPCs", however in the Methods it is described that an equal 50/50 ratio was used of NPCs with these two genotypes. Please clarify if the model was generated with equal ratio or different proportion of these cells.
- 3- In figure 1f, no statistical analysis was associated with it.
- 4- Figure 1i, the canonical pathways show a strong association of the genes analyzed and neurotransmitter and synaptic function. However, the analysis was performed in from RT-PCR array, therefore, the authors should clarify which panel of genes were selected for that assay.
- 5- Figure 3g, the graph shows no statistical differences in the Calcium peak amplitude (as also described in the result section). However, this result is not explained, nor is it explored in the discussion. What is the relevance of the frequency over the amplitude in this context?
- 6- Figure 4k, the improvement described in the result section page 10 "MECP2-KO organoids exhibited decreased population spiking compared to the controls ($P < 0.01$), but treating the MECP2-KO organoids with Nefiracetam and PHA 543613 each improved population spiking ($P > 0.05$ compared to control)" is not statistically significant, therefore, should be described as a trend. Additionally, the actual P value should be specified in the results or in the figure legend.

Referee #1

Trujillo et al. employs human induced pluripotent and embryonic stem cell-derived neurons, neurospheres and cortical organoids models to test the effects of two lead compounds, Nefiracetam and PHA 543613, on rescuing the synaptic defects of Rett syndrome (RTT) caused by mutations of X-linked *MECP2* gene. Although the usage of human stem cell-derived technologies is highly relevant and represents a suitable model for drug screening and selection, there are flaws in the experimental design. The authors show that the two lead compounds can enhance the expression of synaptic markers, yet the effects on network activities are not convincing. In the Discussion, the authors pointed out three intrinsic limitations of this study that were poorly resolved. It is unclear how the two lead compounds may help relieve symptoms of Rett syndrome, and thus limits the clinical relevance of this study. Since the mouse model of Rett Syndrome is widely available from Jackson lab, the authors should test the rescue effects of the two lead compounds in *Mecp2* KO and/or Het mice, at least for several hallmark defects such as synaptic markers level and spine morphology/density.

We thank the reviewer for a careful analysis of our manuscript. We have responded below to all criticisms, which we believe should clarify any concerns and has significantly improved the manuscript.

Major comments:

1. Nefiracetam and PHA 543613 increase expression of synaptic markers Synapsin 1 and PSD-95. Many papers show that deficits of RTT are not only at the level of synaptic proteins. A series of papers show that GABA dysregulation is a major contributing factor in RTT, at least in mice (Chao et al., 2010; Banerjee et al., 2016, Durand et al., 2012). According to existing literatures, GABAergic deficits seem to be more important than glutamatergic neurotransmission. Can the two drug leads proposed by the authors affect inhibition?

We agree that RTT pathophysiology is not restricted to synaptic protein deficiency, but synaptic alteration is central to RTT pathophysiology and protein deficiency has already been used as a primary readout (Banerjee et al, 2019). The reviewer notes that GABAergic alteration has noteworthy contribution to RTT pathophysiology in mice; we selected our pharmacological lead compounds based on the deficiencies that we detected in our *MECP2*-KO human neurons in Figure 1, which broadly included synaptic and neurotransmitter alterations but concentrated in glutamatergic and cholinergic deficits. Nefiracetam is thought to enhance GABAergic, cholinergic, and glutamatergic systems. Nefiracetam shows high affinity for the GABA_A receptor (IC₅₀ = 8.5 nM), where it is presumed to be an agonist (Nabeshima et al., 1990; Goulliaev et al., 1994). On the other hand, PHA 543613 is an $\alpha 7$ -nAChR agonist, and we found no evidence in the literature that it would interact with GABAergic receptors. Moreover, because we do not observe high proportions of GABAergic neurons in our systems (organoids and neurospheres), we presume the effects of the compounds are via their cholinergic and glutamatergic actions.

We now we included additional verbiage in the Discussion that reads (p. 12): “Although pronounced GABAergic pathology has been observed in mice (Chao et al, 2010), we observed lesser GABAergic contribution in our human-based neuronal cultures, and compounds that mainly modulate GABA failed our drug-screening pipeline. We do not observe high proportions of GABAergic neurons in our 3D systems (organoids and neurospheres), so the compounds’ effects that we observed here are presumably via their cholinergic and glutamatergic actions. Nevertheless, inhibitory dysregulation likely contributes to *MECP2*-KO pathophysiology, and future studies of RTT that employ human-based cell models with high GABAergic representation could explore this aspect.” Moreover, we have added in Figure 5 a schematic summary of our model and how the two selected drugs might rescue cellular phenotypes.

2. It provides very limited information when the authors only use synaptic protein level as a major measurement of rescue effects. Other major cellular hallmarks of RTT should be examined. Supple Fig2 describes defects in spine-like morphology such as reduced spine number, spine motility and spine stability. Can the drug leads rescue this defect?

We have clarified in the manuscript that our aim is to use compounds that could interact with neurotransmitter systems in order to promote synaptogenesis and rescue functional network formation. We apologize for the misunderstanding. Moreover, in the manuscript we evaluate synaptic protein content, synaptic density, spine dynamics, calcium activity, and electrophysiologic profile. Now, we have also evaluated the ability of these drugs to rescue nuclei size, morphology, and spine number, but we did not observe any significant rescue. This information is now included in the Results section of the revised manuscript and in Expanded View Figure 5; the additional verbiage reads (p. 10): “Although the primary aim of the drug screening pipeline was to increase synaptogenesis and activity in *MECP2*-KO neurons, morphological features were also investigated. Nefiracetam and PHA 543613 did not increase the number or length of neurites and did not rescue nuclei size (monolayer neurons, neurospheres,

and cortical organoids). However, we observed an increase in neuronal spine-like protrusions due to the treatments (Fig EV5G).”

3. The authors use different readouts to assess network effects in different in vitro models, which make it very difficult to assess the effects.

We agree with the reviewer that many cell types and assays were used in the study and that, although this approach was necessary to rigorously and comprehensively evaluate the effects of the compounds, it can make interpreting the effects of a compound more challenging. Briefly, we selected compounds based on *MECP2*-KO 2D neurons’ neurotransmitter dysregulation, followed by synaptic and functional drug screening. The phenotypic reversal abilities of PHA 543613 and Nefiracetam were then validated in 3D neurodevelopmental models. The overall idea was to find drugs that could rescue *MECP2*-KO phenotypes in a variety of *MECP2*-KO human cell models with heterogeneous levels of network activity. For this reason, our selection of functional assays (synaptogenesis, calcium imaging, and MEA) was particularly based on the cell model being used, allowing us to accurately evaluate the compounds. For these assays, we used 2 different cell lines to mitigate eventual influence of the genetic background on the analysis, and we utilized 3 different stem cell derived systems (neuronal monolayer, neural spheroids, and cortical organoids) to strategically increase the stringency of our screening by adding layers of complexity. However, we recognize that our readers may find it helpful to have a summary representation of our study, and we added this as Figure 5 to summarize these different models and highlight their features as used in this study.

4. Figure 2h,i measures calcium oscillation frequency in stem cell-derived neurons, while Figure 3f and g measures calcium peak frequency and peak amplitude. The definition of calcium peak frequency is unclear and should be described in more detail. Figure 4 i-k uses MEA to measure population spiking in organoids and the rescue effect in Fig 4k is NOT significant.

We apologize for the lack of details; we have now clearly defined all metrics in the Materials and Methods section of the revised manuscript (pgs. 16-18). Rather than calcium oscillation frequency, we have more clearly pointed out that it is calcium transients that we are evaluating in neurons and neurospheres. In addition, we have updated the relevant labels for graphs in Figure 2 and Figure 3.

To speak to the reviewer’s second point, we apologize for the miscommunication regarding Fig 4k (now Fig 4M). We point out in the text that the compounds increase MEA network activity of *MECP2*-KO cortical organoids such that the resultant population spiking activity is not significantly different from the network activity of control organoids. To distinctly highlight this fact for our readers, we added a bar to the graph of Fig 4M that makes it clear that only control and *MECP2*-KO cortical organoids have significantly different network activity from one another.

5. The reviewer suggests the authors to use standardized measurements to assess the individual or network neuronal activities. That is crucial to validate the effects of their two compound leads.

As we noted above, although the many assays and measurements used in this study can make interpretation challenging, doing so was necessary to comprehensively evaluate the compounds relative to one another. All assays used in this study are standards in the field and have been previously published by our lab and others (Marchetto et al. Cell 2010; Chailangkarn et al. Nature 2016; Marchetto et al. Mol Psych 2016; Nageshappa et al. Mol Psych 2016; Trujillo et al. Cell Stem Cell 2019; Sirenko et al. Toxicol Sci 2019). The compounds were only statistically compared to one another within a given assay; conclusions regarding the compounds’ efficacy relative to one another were never drawn between assays. As we note above, we have clarified our assay choices throughout the manuscript, and we added Figure 5 as a schematic summary of the drug screening pipeline.

6. Reduced total RNA has been suggested to be a major cellular deficit in ESC-derived RTT neurons (Li et al., Cell Stem Cell, 2013). Do the two drug leads rescue total RNA level? Could the increase in synaptic markers be a secondary effect of total RNA level rescue?

The reviewer raises an interesting point. We did not observe a reduction of total RNA in our *MECP2*-KO cultures, and our transcriptional analyses (see Figure 4) indicate that treatment increased gene expression in pathways relevant to synaptic function. The lead compounds were screened for their ability to rescue synaptic deficiency specifically in *MECP2*-KO cultures without affecting the controls, indicating the observed treatment effect in the KO cultures is primary, not secondary. We have added verbiage to the Discussion section that reads (p. 14):

“Previous research in human RTT neurons associated MeCP2 deficiency with a reduction in total RNA (Li et al, 2013), suggesting that increased synaptic expression following treatment may be secondary to increased total RNA. However, our transcriptional analyses showed reduced expression of genes concentrated in pathways relevant to synaptic function, and our process of screening synaptically targeted compounds retained only those that increased these deficient pathways in KO cells without affecting controls, suggesting that increased expression of synaptic genes is a primary effect of the targeted therapeutics.”

7. Cortical organoid model is a good model, but the effect of Nefiracetam and PHA 543613 should also be tested in vivo using Mecp2 knockout mice to demonstrate the effect of the two drug leads at the physiological level. The knockout mouse model of Rett Syndrome is widely available from Jackson lab. The authors should test the rescue effects of the two lead compounds in Mecp2 KO and/or Het mice, at least for several hallmark defects such as synaptic markers level and spine morphology/density.

Although we agree with the reviewer that *MECP2*-deficient mice are often used to investigate RTT pathophysiology, we disagree that doing so would offer meaningful contribution to the present manuscript. If these drugs did not work in the Mecp2 knockout mice, it would not mean anything for its translatability. If the drugs work, it will only mean that the system is conserved between the species.

Several recent studies highlight that despite the general conservation, there are extensive differences between homologous human and mouse cell types, including alterations in proportions, distributions, gene expression, and neurotransmission. Notably, the most different systems are glutamatergic, serotonergic, and cholinergic (Hodge et al. Nature 2019; Sjöstedt et al. Science 2020). These species-specific features emphasize the lack of translatability to clinics and the importance of directly studying human brain models. Pharmacologically, the safety of Nefiracetam and PHA 543613 has already been evaluated in animal models, and the safety of Nefiracetam has been demonstrated in human populations (Foucault-Fruchard et al., 2018; Fujimaki et al., 1993; Robinson et al., 2009). In fact, Nefiracetam is already approved for commercial use in humans. Thus, we have included additional verbiage in the Discussion section of our manuscript that reads (p. 13): “Past research has investigated the capacity of pharmacological compounds to rescue RTT pathophysiology using *in vivo* mouse modeling (Tang et al, 2019). Several recent studies highlight that despite general conservation, homologous human and mouse cell types exhibit extensive differences, including alterations in proportion, distribution, gene expression, and neurotransmission. Notably, the most different systems are glutamatergic, serotonergic, and cholinergic (Hodge et al, 2019; Sjostedt et al, 2020). These species-specific features emphasize the lack of clinical translatability and the importance of directly studying human brain models. In addition, as was alluded to above, the safety of Nefiracetam and PHA 543613 have already been evaluated in animal models, and the safety of Nefiracetam has been demonstrated in human populations and is already commercially available for human use (Foucault-Fruchard et al, 2018; Fujimaki et al, 1993; Robinson et al, 2009). We directly demonstrated the effects of these compounds in a human context with a variety of *MECP2*-KO human cell models. Our lab has shown that the most sophisticated of these models, cortical organoids, develop oscillatory activity similar to that observed during human fetal neurodevelopment (Trujillo et al, 2019). Due to the synaptic and network impairment resulting from MeCP2 deficiency, development of similar oscillatory activity likely occurs much later, if at all, in *MECP2*-KO organoids, but documentation and characterization of such activity in future studies may reveal itself as a useful biomarker in a clinical setting.”

8. In Fig 1a and supple Fig 1, the authors show that both iPSC and HESC were used to generate two knockout models (Q83X and K82fs) for the production of neuroprogenitors, neurons, and cortical organoids. However, the particular line used in later figures and legends were not made clear; the authors merely used *MECP2*-KO lines. This is not acceptable since iPSC and HESC are different stem cell models. The authors should specify the line used in each figure for the scientific community to assess the effects of the drug and whether there are line-specific results.

We thank the reviewer for pointing that out. We included in the figure legends where each cell line was used. Additionally, we would like to make clear that neither cell line expressed MeCP2 (*MECP2*-KO lines). We did not observe any significant difference in differentiation efficiency, proliferation, or survival and did not observe any other alteration between Q83X and K82fs cells.

9. It is not useful to compare the size of neurosphere (supple Fig 5) and organoids (Fig 4b) since the size of neurospheres and organoids can vary within and between batches generated at different times. Data on similar organoid size is only shown for wild type (supple Fig 6b). The authors could include a similar graph on knockouts. The authors should compare

individual soma size in their 2D monolayer, neurosphere and organoids models, like supple Fig 2d. and in the author's earlier paper (Marchetto et al., 2010).

We have added the information on the diameter size of *MECP2*-KO cortical organoids to the graph on Expanded View Figure 5. We also included nuclei size analyses for all of these cellular models as suggested (Expanded View Figure 5) and updated the text in the Results section of our revised manuscript such that the pertinent text now reads (p. 10): “Nefiracetam and PHA 543613... did not rescue nuclei size (monolayer neurons, neurospheres, and cortical organoids).”

10. The authors should include more description on how the 14 drug leads were initially selected. Although there is currently no available treatment for RTT, the author should also include a brief discussion on drug leads that are under clinical trials such as IGF-1 and cite the relevant papers.

The drug leads investigated in the present study were selected to counteract the synaptic and neurotransmitter deficiencies that we identified in Figure 1. We enhanced the description and examples with citations (see below) in the Results section of our manuscript, which reads (pgs. 6-7): “We selected 14 pharmacologic compounds with mechanisms of action that counteract the synapse and neurotransmitter pathologies that we identified in Figure 1 (Fig 2B). For instance, because *MECP2*-KO neurons exhibit cholinergic deficiency (Oginsky et al, 2014; Zhang et al, 2016; Zhou et al, 2017), we included compounds that are predicted to specifically promote this action (e.g., Tacrine, Carbamoylcholine).”

To speak to the reviewer's second point, we agree that our readers may appreciate a brief discussion of some of the foremost drug leads that are in or have undergone clinical trial. We have included additional verbiage, including appropriate references and clinicaltrial.gov identification numbers, in our Discussion section that reads (pgs. 11-12): “Several completed and ongoing clinical trials have explored therapeutic safety or efficacy in RTT patients, mainly of IGF-1 or modulators of BDNF, the encoding gene of which is a target of MeCP2 (Chen et al, 2003). Although formulations of IGF-1 are expected to be promising candidates (Pozzo-Miller et al, 2015), results thus far have been mixed. Recombinant human IGF-1 (mescarmin) was safe and mildly efficacious in a phase 1 trial of RTT patients (ClinicalTrials.gov Identifier NCT01253317; Khwaja et al, 2014), but a double-blind, placebo-controlled follow-up study of mescarmin did not show significant phenotypic improvements (NCT01777542; O'Leary et al, 2018). In contrast, trial of a synthetic form of IGF-1 (trofinetide; NCT02715115; Glaze et al, 2017) demonstrated sufficient safety and efficacy to warrant initiation of a phase 3 trial (NCT04279314). Meanwhile, clinical trials that have preliminarily investigated modulators of BDNF have shown efficacy of glatiramer acetate (NCT02153723; Djukic et al, 2016) and led to further trial of fingolimod (NCT02061137).”

11. On page 7, the authors mention that *MECP2*-KO neurons exhibit cholinergic deficiency to justify the use of Nefiracetam and PHA 543613. The authors should cite relevant papers.

As was noted above, we selected compounds with mechanisms of action that counteracted the deficiencies that we had identified in Figure 1. Thus, we included lead compounds that promote cholinergic function because we identified cholinergic deficiency in *MECP2*-KO neurons in the prior figure. Nevertheless, we have cited relevant papers where indicated. The references were added as suggested, including Zhang et al., *Loss of MeCP2 in cholinergic neurons causes part of RTT-like phenotypes via $\alpha 7$ receptor in hippocampus* (2016); Zhou et al., *Selective Preservation of Cholinergic MeCP2 Rescues Specific Rett-syndrome-like Phenotypes in MeCP2 stop Mice* (2017); and Oginsky et al., *Alterations in the cholinergic system of brain stem neurons in a mouse model of Rett syndrome* (2014).

12. The simulation in Fig 2a is interesting, but it does not provide too much information that is clinically relevant to the symptoms of RTT. It is reasonable to predict that rescuing synaptic defects would enhance neuronal network activities. This piece of information aligns with published experimental findings, but is limited in novelty.

Although it is reasonable to predict that rescuing synaptic defects will enhance neuronal network activities, many compensatory factors may influence network function. Synaptic dysfunction is a central pathology of *MECP2* loss of function, so we created and included the model—which highlights the isolated effect of loss/rescue of synaptic puncta on neural activity—as unbiased evidence of the importance of synaptic phenotype. In the ensuing tissue culture experiments, the hypothesis was that changes in synaptic puncta counts drove the corresponding changes in activity, but many other factors are involved; the simulation shows that it is plausible that isolated rescue of synaptic structure, the primary treatment target, is sufficient to rescue network function. Moreover, the simulation supports prediction for the directionality of the change in activity with the change in synaptic puncta. The direct

effect on activity is not obvious in nonlinear recurrent networks. Depending on circuit composition and connectivity, increased or decreased synaptic strengths could each potentially increase or decrease activity. Our simulations show these relationships for different E-I compositions and levels of connectivity, supporting our focus on this mechanism as a therapeutic target.

We have included additional information in the Results section of the manuscript to further underscore for our readers the salience of isolated rescue of synaptic structure, such that the text now reads (p. 6): “Although synaptic pathology is a prominent consequence of MeCP2 deficiency, many factors may influence network function, and it is unclear that targeted treatment of synaptic dysfunction will yield measurably linked improvement in neuronal population activity. Variation between individuals in network connectivity patterns or the proportion of excitatory and inhibitory neurons, for example, may modulate the link between synaptic phenotype and altered neural activity (Van Vreeswijk & Sompolinsky, 1996; Pena et al, 2018). Artificial neural networks offer a biologically plausible framework to explore how parameterized manipulation affects network activity (Kim et al, 2019; Miconi, 2017)...”

13. More description of Supple Fig 6f and g should be included in the legend and in text. The meaning of the figure is not clear.

We agree with the reviewer and decided to replace that figure with more relevant information to the overall study.

14. Methodology can be in more details. Correct grammatical errors.

We have enhanced the Materials and Methods section of our revised manuscript to provide thorough experimental detail and additional citations for experimental methods that we have extensively described in the past. Moreover, we apologize for our grammatical errors, these have been corrected.

15. Since the manuscript contains only 4 main figures. It may be better suited to be reformatted as a short report with 3 figures.

Thank you for the suggestion. We are following *EMBO Mol Med* editorial guidelines.

Referee #2

Trujillo and colleagues investigated the cellular, synaptic, and functional properties of human stem cell-derived neurons, neurospheres, and organoids produced from control and isogenic CRISPR/Cas9-engineered lines with two different MeCP2 mutations. They confirmed synaptic and other deficits in mutant cells and identified promising pharmacological treatments to compensate those deficits. Specifically, they showed that *MECP2*-KO neurons show morphological abnormalities in soma area and spine density, as well as alterations in synaptic gene expression and calcium dynamics. Based on the gene expression data, the authors selected 14 candidate compounds for therapeutic intervention, and then the two most promising ones for further testing. In *MECP2*-mosaic neurospheres, treatments with either drug did not reduce cell viability, but did improve calcium frequency but not amplitude. In *MECP2*-KO organoids that showed decreased size as compared to control organoids, treatments with Nefiracetam or PHA543613 treatment. Based on the findings in this manuscript, the authors draw the conclusion that a pilot trial of Nefiracetam and/or PHA 543613 in treating *MECP2* deficiency is warranted. Overall, the manuscript is strong and the claims are supported by the data. I have only few minor-to-moderate suggestions for improvement:

We thank the reviewer for all the positive comments about the work. We believe our revised version significantly improved with the addition of new data and clearer explanations as suggested.

1. The authors should specify the names and origin of control stem cell lines used in the experiments.

The text was modified accordingly to include this information.

2. The authors should specify throughout the manuscripts which lines and from how many differentiations were used in different experiments, and at which developmental timepoints those experiments were performed. For example, in Fig. 1, it is unclear which lines were used for PCR and immunostaining assays and when those assays were performed (Fig. 1g-h). In addition, the authors should describe why those primers were selected. It is surprising that only few excitatory synaptic markers were tested in those experiments. Accordingly, the claim "Many of the implicated genes appeared

relevant to synaptic function, and the differences in expression became increasingly pronounced as differentiation progressed" (page 5), does not seem to be supported by the data and should be revised.

The text was modified accordingly to include this information. We provided these details in the Materials and Methods, Figure legends, and Results. The sentence noted by the reviewer was modified to read (p. 5): "Many of the implicated genes appeared relevant to synaptic function and showed differences in expression during differentiation."

3. It is unclear what experiments were performed to characterize cell-type specific marker expression in Fig 1h. I assume it is immunostaining; however, it is not stated either in the text or in the figure legend. If it is immunostaining, the authors should present some representative images supporting the quantifications. In addition, the authors should discuss why reduced proportions of layer V and VI cortical neurons were detected in *MECP2*-KO and how this result might affect synaptic maturation and connectivity.

We apologize for the lack of clarity. The experiment performed was single-cell qPCR gene expression analysis (Fluidigm). We have included in Expanded View Figure 2 some immunostaining images of some of the markers that were used to characterize the neuronal populations. Likewise, the manuscript text was updated with this information and pertinent supporting references were added.

Regarding the reviewer's second point, we agree that the reduced proportion of layer V/VI cortical neurons in *MECP2*-KO monolayer neuronal cultures is an interesting phenomenon. The mechanism by which this occurs is unclear, but layer V/VI neurons are known to be heterogeneous as a class in their morphological and electrophysiological properties, gene expression, and axonal projections, and *MeCP2* may be hypothesized to affect any of these processes during neurodevelopment (Stuss et al., 2012). The effects of *MeCP2* mutations are also thought to vary considerably between contexts and brain regions (Stuss et al., 2012). Interestingly, in our cultures, although the proportion of CTIP2+ cells was significantly decreased in *MECP2*-KO monolayer neurons, a slighter decrease was observed in *MECP2*-KO cortical organoids that did not reach significance. To bring these points to our readers' attention, we have included additional verbiage in the Discussion section of our manuscript that reads (p. 16): "It is unclear why populations of *MECP2*-KO monolayer neurons showed a smaller proportion of layer V/VI neurons, but it has been noted that these neurons are profoundly heterogeneous in their morphology, gene expression, axonal projection, and electrophysiological function and that *MECP2* may affect any of these parameters in a context- and region-specific manner (Stuss et al., 2012). It would be reasonable to expect that a smaller proportion of these neurons would reduce synaptic maturation and connectivity, but correcting the mechanism responsible for this deficit plainly requires compounds whose function is not restricted to the synapse and may require intervention at an earlier timepoint."

4. The significance of GO analysis in Fig. 1 is unclear. Only few selected genes were tested in PCR experiments, and the panel of excitatory synaptic genes seems to be incomplete.

The text was modified accordingly to include this information. We added Appendix Table 1 with all 96 single-cell qPCR and 86 qPCR Array markers. Having identified altered expression of synaptic genes via the qPCR experiments in the preceding panels of Figure 1, we included the GO analysis as a summary showing that alterations primarily concentrate in particular neurotransmitter synthesis and receptor pathways in confirmatory agreement with our quantification. We updated our Results to more clearly point this out to our readers; the text now reads (p. 5): "Transcriptional analysis of genes related to neurotransmission in *MECP2*-KO populations showed changes in glutamatergic, GABAergic, and cholinergic systems ($P < 0.0001$; Fig. 1H, bottom, and Fig EV3G-K) A summary gene ontology analysis confirmed these findings, showing that alterations in gene expression due to *MECP2*-KO primarily concentrated in particular neurotransmitter synthesis and receptor pathways in overlapping alignment with the results of our quantitative analysis (Fig 1I)."

5. Authors should provide more details on the differentiation protocols used in this study. For example, it is unclear for how many days neural progenitors were treated with FGF2.

We apologize for the lack of details. We have included more information in the Methods section. We have updated our Methods so that how we are performing neuronal differentiation is clearer for our readers. The relevant portion of the text now reads (p. 18), "Emergent NPCs were expanded and maintained in N2B27 medium with feeding on alternate days; all NPCs used for neurons were passage 5-20. FGF-2 was withdrawn from the medium to induce neuronal differentiation, considered Day 0 of differentiation."

6. The rationale and value of network modeling is unclear. It is expected that increased synaptic transmission should lead to increased probability of neuronal spiking. Can this model predict the extent to which synaptic transmission should be increased to improve spiking but not to induce seizures?

Synaptic dysfunction is a central pathology of *MECP2* loss of function, so we created and included the model as unbiased evidence of the importance of synaptic phenotype. In the ensuing tissue culture experiments, the hypothesis was that changes in synaptic puncta counts drove the corresponding changes in activity, but many other factors are involved; the simulation shows that it is plausible that isolated rescue of synaptic structure is sufficient to rescue network function. The simulation, moreover, supports prediction for the directionality of the change in activity with the change in synaptic puncta. The direction of effect on activity is not obvious in nonlinear recurrent networks. Depending on circuit composition and connectivity, increased or decreased synaptic strengths could each potentially increase or decrease activity. Our simulations show these relationships for different E-I compositions and levels of connectivity, supporting our focus on this mechanism as a therapeutic target.

The reviewer asks an interesting question about predicting optimal increase in synaptic transmission to alleviate symptoms while not inducing seizures. Although the model may be used to predict the approximate extent to which synaptic transmission should be increased to improve spiking, it was not designed to study seizure phenotypes; however, development of a model to predict activity patterns leading to seizures and calibrating it to experimental findings would be a promising line of future research.

To clarify these points for readers, we updated the Results section of our manuscript such that it now reads (pgs. 6): “Although synaptic pathology is a prominent consequence of *MeCP2* deficiency (Gonzales & LaSalle, 2010; Nguyen et al, 2012; Johnston et al, 2003), many factors may influence network function, and it is unclear that targeted treatment of synaptic dysfunction will yield measurably linked improvement in neuronal population activity. Variation between individuals in network connectivity patterns or the proportion of excitatory and inhibitory neurons, for example, may modulate the link between synaptic phenotype and altered neural activity (Van Vreeswijk & Sompolinsky, 1996; Pena et al, 2018). Artificial neural networks offer a biologically plausible framework to explore how parameterized manipulation affects network activity (Kim et al, 2019; Miconi, 2017)...”

7. The authors used synaptic staining as a primary assay to detect and rescue synaptic deficits. It is important to know the robustness of this assay. The authors should calculate and report Z-factor (Zhang, Chung, and Oldenburg J Biomol Screen. 1999).

This information was added to the Figure legends alongside the statistical presentations.

8. The authors should provide more details on the synaptic assay (Fig. 2). Which dendrites (proximal or distal) were selected for analysis? How many dendrites were analyzed? Has this been done manually or automatically by a blinded or non-blinded investigator?

We have updated our Methods section with more details, and the text now reads (p. 27): “...a blinded investigator manually quantified co-localized synaptic puncta along 50 μm segments of randomly selected MAP2-positive processes (7-8 per condition); for quantification, no distinction was made between proximal and distal segments.”

9. The authors should specify when different experiments on organoids were performed. For example, they report that ~80% of cells in organoids expressed NeuN. NeuN is a mature neuronal marker. When was this measured?

We apologize for this oversight. The cortical organoids were measured at 2-3 months. The data agrees with Trujillo et al. Cell Stem Cell, 2019. We have amended the text to include this information, and the relevant portion of the Results now reads (p. 9): “One-month old cortical organoids were treated for another month with either Nefiracetam (1 μM) or PHA 543613 (1 μM), and experiments and analyses were performed using organoids at 2-3 months of age (Trujillo et al, 2019).”

10. Untreated organoids in Fig 4d and Fig. 4e show drastically different gene expression profiles. Why is that? The authors should specifically discuss the similarity and differences in expression networks altered by Nefiracetam and PHA543613 treatments. In addition, the authors should present gene expression data obtained in isogenic control organoids. Otherwise, it is unclear whether the treatments rescue gene expression profiles.

Thank you to the reviewer for pointing that out. We would like to clarify that we used different cell lines for the heatmap representation. We have now included genes and markers to clearly specify the drug-induced gene expression changes. Additionally, we have compared the gene expression of both untreated *MECP2*-KO cell lines. You can appreciate below that although both cell lines show differences of relative gene expression (heatmap), the

overall impact in the neurogenesis is mild. The GO shows several not significant changes and terms that are not relevant to the analysis.

11. The authors should provide high-magnification images of synapses in Fig. 4f. The presented images are of poor quality. They should also discuss how synapses were quantified in organoid sections. Where and how the ROIs were selected.

The figure was updated. The complete analysis is also included in Expanded View Figure 5.

12. In the previous study, the authors investigated network oscillations in stem cell-derived organoids. It would be to show the comparison of oscillations in control and MeCP2-deficient organoids. This might serve as a biomarker in the future clinical studies.

The reviewer makes an interesting observation. Unfortunately, performing such an analysis even in control cortical organoids requires organoids of at least six months of age. Considering the synaptic and network impairment resulting from MECP2 deficiency, developing oscillatory activity in MECP2-KO organoids that is sufficiently robust to accurately perform this analysis will likely require many more months. Nonetheless, we now included additional verbiage in our Discussion section that reads (pgs. 13-14): “Our lab has shown that the most sophisticated of these models, cortical organoids, develop oscillatory activity similar to that observed during human fetal neurodevelopment (Trujillo et al., 2019). Due to the synaptic and network impairment resulting from MeCP2 deficiency, development of similar oscillatory activity likely occurs much later, if at all, in MECP2-KO organoids and was not observed in the present study, but documentation and characterization of such activity in future studies may reveal itself to be a useful biomarker in a clinical setting.”

13. The authors should discuss why Nef and PHA treatment do not rescue network deficits in organoids (Fig 4k shows p value > 0.05).

The most likely reason that Nefiracetam and PHA 543613 do not rescue network deficits in cortical organoids to the full activity level of control organoids is that network activity is the summary result of many influences, synaptic and otherwise, and the synaptic deficiencies may extend beyond those combatted by our selected compounds’ mechanisms of action. We have modified the Results and Discussion sections of our manuscript to make these details more easily accessible for readers. In the Results section, the text now reads (p. 10): “MECP2-KO organoids exhibited decreased population spiking compared to the controls ($P < 0.01$), but treating the MECP2-KO organoids with Nefiracetam and PHA 543613 each increased population spiking to a level approaching that of the controls (Nefiracetam, $P = 0.98$ compared to control; PHA 543613, $P = 0.12$ compared to control; Fig 4K-M).” We have also enhanced the limitations section of our Discussion to more explicitly note this fact, and the text now reads (p. 14): “A third limitation is that the compounds we isolated, Nefiracetam and PHA 543613, were unable to fully reverse all of the MECP2-KO phenotypes we identified. Although Nefiracetam and PHA 543613 increased the MEA network activity of MECP2-KO cortical organoids, they did not fully restore activity, suggesting other deficiencies also contribute to MECP2-KO network pathology, synaptic and otherwise, that must be investigated and targeted with other compounds... Treatment of diseases incident to MeCP2

deficiency may thus be more complete with a combination of drugs, another reason to explore a wider array of compounds.”

14. The sample size should be stated for all quantifications including those in Fig. 1.

The sample size has now been included in the figure legends.

15. Scale bars are not defined or missing on most of the images.

We included the missing error bars accordingly.

16. Figure S3b: Text is unreadable.

The Figure was modified for better readability.

Referee #3

In the present work, the authors take advantage of previously observed phenotypes associated with *MECP2*-KO neurons, such as synaptic formation and neuronal maturation, to assess pharmacological compounds able to (partially) reverse those phenotypes. The authors performed an extensive array of assessments of neurons and organoids, from genomic to morphological and functional approaches. One very elegant aspect of the present study was the use of both 2D and 3D human stem cell models. In total, there were 14 drugs selected based upon their mechanism of action, targeting different synaptic proteins. From those, Nefiracetam and PHA 543613, specifically partially rescued synaptogenesis defects, calcium signaling, and impaired neural network activity. The manuscript is well written and organized and the data is presented in a clear fashion. However, it is important to highlight that, while the overall story and subject are relevant, there are concerns regarding *MECP2*-mosaic neurospheres and the selection of the final drugs that should be addressed.

We thank the Reviewer for critical reading of our manuscript for all the positive comments.

Major Concerns:

1. The authors initially selected 14 compounds based on their mechanisms of action associated with synapsis and neurotransmitters. However, the mechanism connecting the drugs, especially Nefiracetam and PHA 543613, to *MECP2* was not investigated in this work. That information would be important to assess if the efficiency of the drugs are specific to *MECP2*-KO pathogenesis and should be better explored in the discussion.

We thank the reviewer for pointing that out. We included an improved model on the specific effects of these drugs on the *MECP2*-KO pathogenesis. You can find this information in Figure 5. However, we explained that we cannot rule out the possibility of an indirect effect of the drugs or resulting changes in gene expression. We also enhanced the Discussion section of our manuscript to better highlight for our readers how we think the drug compounds are exerting their effects. Including some of the prior verbiage, the text now reads (p. 11-12): “Our *MECP2*-KO neuronal cultures showed that synaptic and neurotransmitter pathophysiology principally concentrated in glutamatergic and cholinergic dysregulation. Clinically, the lead compounds Nefiracetam and PHA 543613, both of which are orally administered, have invaluable mechanisms of action to treat these deficiencies. Nefiracetam is a cholinergic, GABAergic, and glutamatergic agonist developed to enhance cognitive functioning (Moriguchi, 2011; Malykh & Sadaie, 2010), and PHA 543613 is an $\alpha 7$ -nAChR agonist with proven neuroprotective effects in a neurodevelopmental disease model (Foucault-Fruchard *et al*, 2018). Cholinergic modulatory effects within the nervous system are many because nAChRs are widely dispersed across the neuronal and synaptic architecture, and acetylcholine additionally affects the release of other neurotransmitters (Picciotto *et al*, 2012).”

2. A very elegant addition to the work of the *MECP2*-mosaic model, mimicking the female RTT syndrome brain. However, no rationale was given as to why the authors chose to mix NPCs and work with neurospheres. A more relevant model would be the cortical organoids which could be achieved by mixing the iPSCs. In the same context, calcium influx was used as a readout for the *MECP2*-mosaic model. Calcium activity was used as a proxy for the neurotransmitter's defects; however, it would be crucial to assess if their electrophysiology potential is different and if that could also be rescued using Nefiracetam and PHA 543613 or the combination of those drugs.

We agree with the reviewer that a cortical organoid model of RTT mosaicism would be fascinating. Unfortunately, the methodology we use to generate cortical organoids precludes accurately performing this analysis. Our cortical organoids start as single pluripotent stem cells that self-aggregate in suspension in a 6-well plate. Although

different proportions of control and *MECP2*-KO stem cells could be combined in a particular well, controlling the proportions of *MECP2*-KO and control cells in a resultant self-aggregated organoid is impossible, and statistically adjusting for this after-the-fact, as another option, would demand an extreme sample size to undertake appropriate analysis and would be much less accurate than the more direct approach of using neurospheres that we selected. To point this out to our readers, we have included additional verbiage in our Discussion that reads (p. 15): “Although a cortical organoid model of RTT mosaicism could potentially have yielded additional findings, the methodology used here to generate cortical organoids (single PSCs self-aggregating in suspension) precludes accurately performing this analysis, as it would be impossible to control the final proportions of control and *MECP2*-KO cells in a particular organoid. We elected instead to model mosaicism using the more direct approach of neurospheres so that we could minimize variability between samples and maximize our ability to accurately discern the effects of the compounds. Nevertheless, development of a cortical organoid model of RTT mosaicism presents an interesting future direction for study.”

Regarding the reviewer’s second point, we performed MEA recordings on the *MECP2*-mosaic models, untreated and treated with the selected drugs. The results of these assays are now displayed in Figure 3, and in the Results section of the manuscript, which reads (p. 9): “...we investigated the impact of the drug treatments on the electrophysiological properties of the *MECP2*-mosaic model using MEA analysis. We observed a trend of decreased spike frequency as the percentage of *MECP2*-KO cells in the neurospheres increased. Despite variability in the recordings, both PHA 543613 and Nefiracetam increased the overall spike count after 5 weeks of treatment (Fig 3J and K).”

3. The majority of the experiments that assessed the efficacy of the drugs only displayed a partial rescue, including the electrophysiology of the cortical organoids, therefore, the text should be edited to properly represent that. I.e. Introduction: "Two currently available lead compounds, Nefiracetam and PHA 543613, exhibited therapeutic potential to rescue the synaptic and functional network defects caused by MeCP2 deficiency and are viable candidates for clinical trial" (page 4) should read "...exhibited therapeutic potential to partially rescue the synaptic and functional network."

The text was modified accordingly.

Minor Concerns

1. It was not clear in the results or methods' sections how were the concentrations of the drugs selected. Was there a concentration curve for these drugs other than in figure 3? The same question could be applied for the selected time points presented in this work.

The concentration of the drugs was selected based on the specific K_d reported for each drug and from previous work on animal models and human neurons. The time points were strategically selected after we observed a cellular or functional phenotype in *MECP2*-KO neurons. Based on published work from our group, *MECP2*-KO neurons and neurospheres already show a distinct deficit after 4 weeks. Although we observed that acute or 1 week of treatment did not improve synaptogenesis, treatment longer than 2 weeks was able to rescue some phenotypes. The Materials and Methods section of our manuscript was modified to include this information, and the text now reads (p. 19): “One-month old cortical organoids were treated for one month with either Nefiracetam (1 μ M) or PHA 543613 (1 μ M) during every-other-day media changes. The concentration of the drugs was selected based on the specific K_d reported for each drug and from previous work on animal models and human neurons. The time points were strategically selected after we observed a cellular or functional phenotype in *MECP2*-KO neurons. Based on published work from our group, *MECP2*-KO neurons and neurospheres already show a distinct deficit after 4 weeks.”

2. In the result section entitled: "Treatment of *MECP2*-mosaic neurospheres increases cell viability and calcium activity" the authors describe the *MECP2*-mosaic as a combination of "different proportions of control and *MECP2*-KO NPCs", however in the Methods it is described that an equal 50/50 ratio was used of NPCs with these two genotypes. Please clarify if the model was generated with equal ratio or different proportion of these cells.

We have updated our text such that it now reads (p. 8): “...we developed a model of *MECP2* cellular mosaicism (*MECP2*-mosaic) to mimic the female RTT brain by combining control and *MECP2*-KO NPCs in a 50/50 mixture.”

3. In figure 1f, no statistical analysis was associated with it.

We apologize for the misunderstanding. We used the Violin Plot only to visualize the distribution of the data and its kernel probability density. However, the adjacent Volcano plot shows statistical significance versus fold change to enable quick visual identification of genes with large fold changes and significance (Fig 1G).

4. Figure 1i, the canonical pathways show a strong association of the genes analyzed and neurotransmitter and synaptic function. However, the analysis was performed in from RT-PCR array, therefore, the authors should clarify which panel of genes were selected for that assay.

We have included Appendix Table 1 showing all single-cell qPCR and qPCR Array markers used.

5. Figure 3g, the graph shows no statistical differences in the Calcium peak amplitude (as also described in the result section). However, this result is not explained, nor is it explored in the discussion. What is the relevance of the frequency over the amplitude in this context?

The reviewer introduces an interesting point. Because the effects of the compounds and the clinical implications of any given assay are challenging to predict, we sought to comprehensively investigate and present an array of parameters. Our findings that the two compounds increase calcium transient frequency in *MECP2*-mosaic neurospheres align with our findings in the previous figure (Figure 2H), where we observed that Nefiracetam and PHA 543613 increased calcium frequency and activity in *MECP2*-KO neurons. The concordance of these findings between assays and models affirms our conclusion that the compounds are meaningfully improving deficits associated with MeCP2 deficiency. To draw our readers' attention to this point, we have included additional verbiage in the limitations section of our Discussion, which reads (p. 15): "In addition, neither compound significantly increased calcium transient amplitude in *MECP2*-mosaic neurospheres, but the compounds did increase their calcium transient frequency, a finding that aligns with our observation that Nefiracetam and PHA 543613 increased calcium transient frequency and activity in *MECP2*-KO neurons. The clinical implication of calcium frequency vs amplitude in *MECP2*-mosaic neurospheres is unclear and presents an important direction for future study, but the concordance of our findings between assays and models affirms our confidence in the compounds' effects."

6. Figure 4k, the improvement described in the result section page 10 "*MECP2*-KO organoids exhibited decreased population spiking compared to the controls ($P < 0.01$), but treating the *MECP2*-KO organoids with Nefiracetam and PHA 543613 each improved population spiking ($P > 0.05$ compared to control)" is not statistically significant, therefore, should be described as a trend. Additionally, the actual P value should be specified in the results or in the figure legend.

We have modified the Results and Discussion sections of our manuscript to make these details more easily accessible for our readers. In the Results section, the text now reads (p. 10): "*MECP2*-KO organoids exhibited decreased population spiking compared to the controls ($P < 0.01$), but treating the *MECP2*-KO organoids with Nefiracetam and PHA 543613 each increased population spiking to a level not significantly different from that of control organoids (Nefiracetam, $P = 0.98$ in comparison to control organoids; PHA 543613, $P = 0.12$ in comparison to control organoids; Fig 4K-M)." We have also enhanced the limitations section of our Discussion to more explicitly note this fact, and the text now reads (p. 14): "A third limitation is that the compounds we isolated, Nefiracetam and PHA 543613, were unable to fully reverse all of the *MECP2*-KO phenotypes we identified. Although Nefiracetam and PHA 543613 increased the MEA network activity of *MECP2*-KO cortical organoids, they did not fully restore activity, suggesting other deficiencies also contribute to *MECP2*-KO network pathology, synaptic and otherwise, that must be investigated and targeted with other compounds... Treatment of diseases incident to MeCP2 deficiency may thus be more complete with a combination of drugs, another reason to explore a wider array of compounds." Note that we included the actual P value both in the Results section of the manuscript and in the legend for Figure 4 as advised.

30th Sep 2020

Dear Dr. Muotri,

Thank you for the submission of your revised manuscript to EMBO Molecular Medicine. We have now received the enclosed reports from the referees that were asked to re-assess it. As you will see the reviewers is now globally supportive and I am pleased to inform you that we will be able to accept your manuscript pending the following final amendments:

***** Reviewer's comments *****

Referee #1 (Remarks for Author):

The authors revised the manuscript significantly. The reviewer still think that mouse study will provide insights on the cell-type specific actions of the drug leads. However, the reviewer agrees that mouse study may be outside the scope of this current manuscript. Enhanced discussion and method section is now included.

Referee #2 (Remarks for Author):

The authors has revised the manuscript according to my suggestions. I recommend it for publication.

REVIEWER'S COMMENTS

Referee #1 (Remarks for Author):

The authors revised the manuscript significantly. The reviewer still think that mouse study will provide insights on the cell-type specific actions of the drug leads. However, the reviewer agrees that mouse study may be outside the scope of this current manuscript. Enhanced discussion and method section is now included.

We appreciate the reviewers time and effort to improve our study.

Referee #2 (Remarks for Author):

The authors has revised the manuscript according to my suggestions. I recommend it for publication.

We are glad that our revisions met your standards. We appreciate your time and criticism on our study.

The authors performed the requested changes.

YOU MUST COMPLETE ALL CELLS WITH A PINK BACKGROUND ↓
PLEASE NOTE THAT THIS CHECKLIST WILL BE PUBLISHED ALONGSIDE YOUR PAPER

Corresponding Author Name: Cleber A. Trujillo and Alysson R. Muotri

Manuscript Number: EMM-2020-12523-T